# Functional regulation of an ancestral RAG transposon *ProtoRAG* by a *trans*-acting factor YY1 in lancelet

Song Liu [1,4], Shaochun Yuan [1,2,4✉], Xiaoman Gao[1], Xin Tao[1], Wenjuan Yu[1], Xu Li[1], Shangwu Chen[1] & Anlong Xu [1,3✉]

The discovery of ancestral RAG transposons in early deuterostomia reveals the origin of vertebrate V(D)J recombination. Here, we analyze the functional regulation of a RAG transposon, *ProtoRAG*, in lancelet. We find that a specific interaction between the *cis*-acting element within the TIR sequences of *ProtoRAG* and a *trans*-acting factor, lancelet YY1-like (bbYY1), is important for the transcriptional regulation of lancelet *RAG*-like genes (*bbRAG1L* and *bbRAG2L*). Mechanistically, bbYY1 suppresses the transposition of *ProtoRAG*; meanwhile, bbYY1 promotes host DNA rejoins (HDJ) and TIR-TIR joints (TTJ) after TIR-dependent excision by facilitating the binding of bbRAG1L/2 L to TIR-containing DNA, and by interacting with the bbRAG1L/2 L complex. Our data thus suggest that bbYY1 has dual functions in fine-tuning the activity of *ProtoRAG* and maintaining the genome stability of the host.

[1] State Key Laboratory of Biocontrol, Guangdong Province Key Laboratory of Pharmaceutical Functional Genes, School of Life Sciences, Sun Yat-sen University, 510275 Guangzhou, People's Republic of China. [2] Laboratory for Marine Biology and Biotechnology, Qingdao National Laboratory for Marine Science and Technology, 266237 Qingdao, People's Republic of China. [3] School of Life Sciences, Beijing University of Chinese Medicine, 100029 Beijing, People's Republic of China. [4]These authors contributed equally: Song Liu, Shaochun Yuan. ✉email: yuanshch@mail.sysu.edu.cn; lssxal@mail.sysu.edu.cn

The emergence of recombination activating gene (RAG) is considered a milestone event in the genesis of the adaptive immune system of jawed vertebrates[1]. Guided by the well-known 12/23 rule, jawed vertebrate RAG machinery can mediate V(D)J recombination to produce highly diversified antigen receptors[2,3]. The inverted pairing of 12-RSS (recombination signal sequence) and 23-RSS was reminiscent of terminal inverted repeats (TIRs) flanked at both ends of a DNA transposon. Similarity between V(D)J recombination and the cut-and-paste DNA transposition in their early steps led to the hypothesis that a RAG transposon composed of adjacent RAG1 and RAG2 genes flanked by RSS-like TIRs was the source of jawed vertebrate RAG genes and the origin of split antigen receptor genes[4]. Several lines of biochemistry evidence have been reported since the RAG transposon hypothesis was proposed in the late 1970s[5], and direct evidence supporting such a RAG transposon hypothesis was provided by the discovery of an active ProtoRAG transposon from the lancelet germline[6].

A typical ProtoRAG transposon contains a pair of tail-to-tail-oriented RAG1-like and RAG2-like genes, which are flanked by 5-bp target site duplications (TSDs) and a pair of TIRs. Similar to RSS, which contains a characteristic heptamer and nonamer, TIRs of ProtoRAG also contain a conserved 7-bp element (5′-CAC-TATG-3′) known as the TIR region 1 (TR1) and a conserved 9-bp element (5′-GCCATCTTG-3′) named TR5[7,8]. The sequence of TR1 is similar to that of the RSS heptamer (5′-CACAGTG-3′), while the sequence of TR5 is quite different from that of the RSS nonamer (5′-ACAAAAACC-3′). The lancelet RAG1L/RAG2L protein complex can mediate TIR-dependent transposon excision, host DNA rejoining (HDJ), transposition, and even signal joint (SJ) formation at a low frequency, as in the case of jawed-vertebrate RAGs[6].

Since transposition events resulting from an active DNA transposon may lead to gene deletion, inversion, and even genomic instability, the host has to develop mechanisms to suppress the activities of DNA transposons during evolution, leading to the fossilization of almost all DNA transposons in mammalian cells[9,10]. For example, the host can restrict its transposition sites or reduce the expression level and activities of a P element transposase from germline to somatic cells[11,12]. An RNA-binding protein Hfq may negatively regulate the transposition of Tn5 and Tn10 transposons by suppressing the expression of their IS50 and IS10 transposases at the post-transcriptional level[13,14]. The vertebrate RAGs are also restricted by trans and cis elements to inhibit their endonuclease activity[15–17]. Following the identification of ProtoRAG in the basal chordate Branchiostoma belcheri, the RAG transposon was also predicted in the hemichordate Ptychodera flava, supporting the possibility that the ancestral RAG transposon remains active in several deuterostome lineages[18]. However, although approximately 53 ProtoRAG transposon copies have been identified by scrutinizing the available data for the lancelet genome, most of these copies were not intact[6]. Only three polymorphic insertions of ProtoRAG were identified in the lancelet genome in a previous study[6]. Other studies on ancient RAG transposons have also suggested that most of the predicted RAG transposons should have been fossilized and domesticated to be host genes, at least in the case of jawed vertebrates[6,7,18]. Thus, unraveling the regulatory mechanisms of ProtoRAG in lancelet will help us understand how the host restricted the activity of ProtoRAG, driving its domestication to host genes.

Here, we demonstrate that a trans-acting factor, lancelet YY1-like (bbYY1), and the related cis-acting element within the TIR sequences of ProtoRAG are important for the transcriptional regulation of lancelet RAG-like genes. Meanwhile, bbYY1 suppresses the transposition of ProtoRAG but benefits the host DNA rejoins (HDJ) and TIR-TIR joints (TTJ) after TIR-dependent excision. Moreover, bbYY1 interacts with the bbRAG1L/2 L complex and facilitates the precise binding of bbRAG1L/2L to the TIR-containing DNA. These results suggest that bbYY1 acts as a double-edged sword that finetunes the activity of ProtoRAG and maintains the genome stability of the host. These findings may help us understand the rationale of the long-term survival of these ancestral RAG transposons Proto-RAG and shed new light on the correspondence between TE regulation and genomic stability.

## Results

**ProtoRAG is self-activated by its flanking TIR elements.** Long terminal repeats (LTRs) or TIRs of transposon elements (TEs) usually self-activate or silence the transcription of their encoded transposases[19–21]. To examine whether ProtoRAG is a self-activated DNA TE, we cloned the ~1800-bp sequence upstream of bbRAG1L translation start site "ATG" and the ~600-bp sequence upstream of bbRAG2L "ATG" and then inserted them into the pGL$_3$-basic luciferase reporter construct to determine whether there were cis-acting elements within ProtoRAG to drive the transcription of its encoded bbRAG1L and bbRAG2L genes (Fig. 1a). In our previous study, the minimal TIR sequences needed for ProtoRAG transposon excision were determined to contain the first 43 bp of a 5′ TIR paired with the first 47 bp of a 3′ TIR. This minimal TIR pair was shown to mediate ProtoRAG transposon excision efficiently with low background noise[6]. Thus, for clarity, in this study, the full-length TIR sequences were indicated as 5′/3′ TIR-FL, while the core minimal TIR pair was indicated simply as 5′/3′ TIR. The luciferase reporter results showed that both 5′ TIR-FL and 3′ TIR-FL of ProtoRAG, but not their flanking sequences, contained cis-acting elements that could activate the transcription of reporter genes (Fig. 1a, b). The transcriptional activity of ProtoRAG was comparable to that of the SV40 promoter but lower than that of the CMV promoter (Fig. 1d). Since we could not identify the transcription start sites (TSSs) of bbRAG1L and bbRAG2L by rapid amplification of cDNA ends (RACE) from cDNA of adult lancelet due to their extremely low expression, we had to transfect the luciferase reporter constructs pGL-R1-1 and pGL-R2-1 into 293T cells to obtain the TSS of the firefly luciferase gene to predict the potential TSSs of bbRAG1L and bbRAG2L. The results showed that the potential TSS of bbRAG1L was located near the 5′ TIR-FL and that of bbRAG2L was located within the 3′ TIR-FL (Fig. 1a, i). Similar to TIRs on a common DNA transposon or LTRs on a retrotransposon[22], the reporter assays showed that both cis-acting elements within 5′ and 3′ TIR-FLs had bidirectional transcription activities (Fig. 1e). Then, a construct named pdTIR, in which the luciferase gene was flanked by a pair of TIR-FLs, was used for further reporter assays (Fig. 1f). The results showed that the cis-acting element in the 3′ TIR-FL could enhance the transcriptional activity of that in the 5′ TIR-FL (Fig. 1c).

To further determine the core activating elements, several truncated TIR-FL sequences of ProtoRAG were cloned and inserted into the pGL$_3$-basic vector. The results showed that the region −288 to −268-bp upstream of the bbRAG1L potential TSS and the region +17 to +65-bp downstream of the bbRAG2L potential TSS were identified as the mini-core elements for the transcription of bbRAG1L and bbRAG2L, respectively (Supplementary Fig. 1A–D). Both mini-core elements were within the region of the 5′ TIR-FL and 3′ TIR-FL (Supplementary Fig. 1E). Taken together, these findings suggested that ProtoRAG was a TIR-dependent and self-activated DNA transposon with 5′ and 3′ TIR-FL sequences that may coordinate with each other to drive the transcription of its encoded bbRAG1L and bbRAG2L transposases.

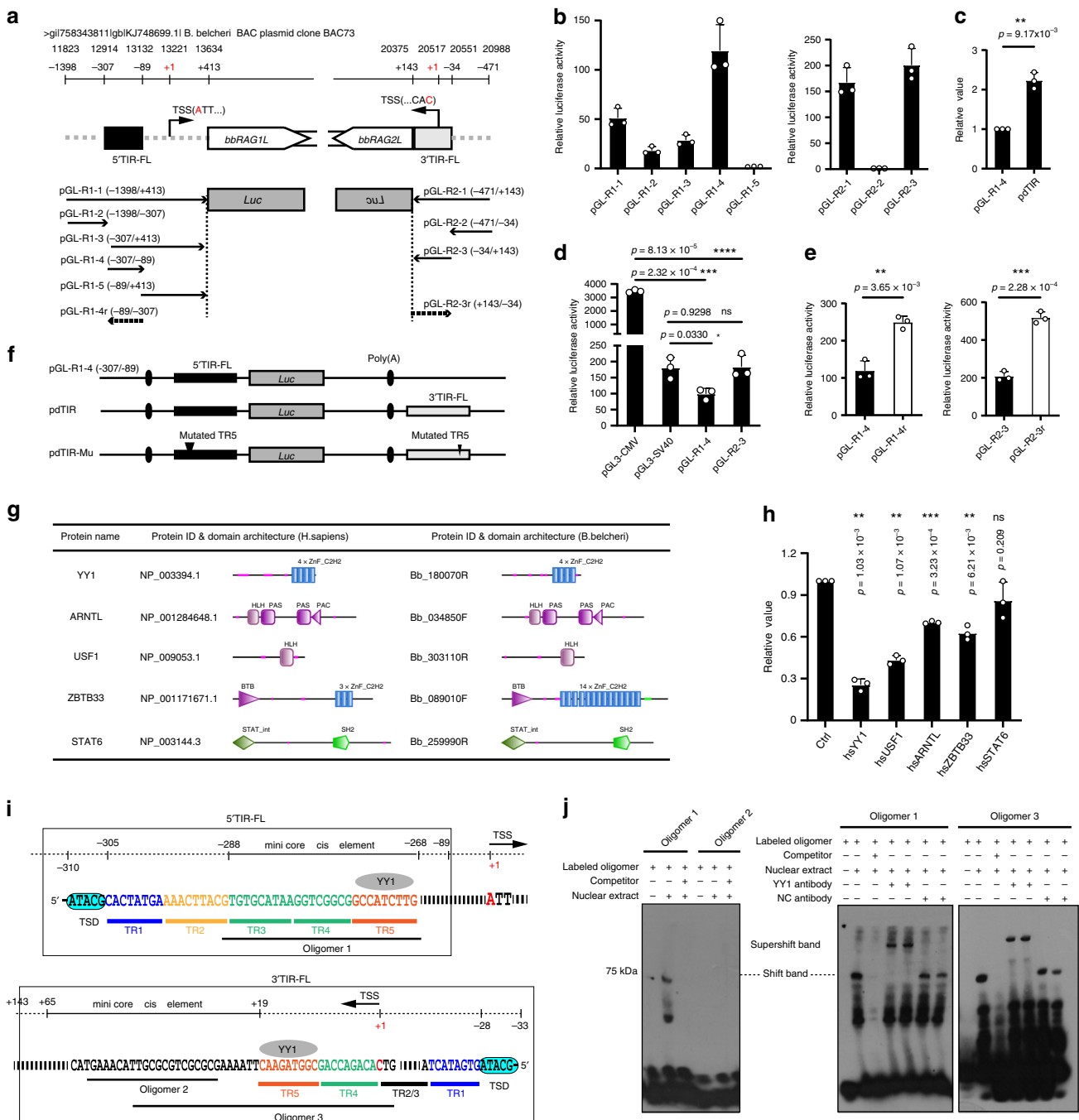

**Fig. 1 Identification of *cis*-acting elements and YY1-binding motif of *ProtoRAG*. a** Genomic architecture of *ProtoRAG* on BAC73 clone from *B. belcheri* BAC libraries and constructs generated for luciferase reporter assay. Potential TSSs are indicated in red. **b** Luciferase reporter experiments showed that both full-length 5′ and 3′ TIRs (TIR-FL) of *ProtoRAG* contain *cis*-acting elements that may drive the transcription of bbRAG1L and bbRAG2L. See also Supplementary Fig. 1A–D. **c** The 3′ TIR-FL of *ProtoRAG* can enhance the transcriptional activity of 5′ TIR-FL by luciferase reporter experiments. **d** The luciferase reporter assay indicated that the transcriptional activity of TIR-FL is comparable to that from the *SV40* promoter but lower than that from the *CMV* promoter. **e** *Cis*-acting elements in both 5′ and 3′ TIR-FLs have bidirectional transcription activities according to the analysis of data from luciferase reporter assays. The reporter assays were performed in 293T cells with TIR-containing plasmids, as shown in **a**. **f** Schematic diagram of pdTIR and pdTIR-Mu constructs. pTIR-Mu indicates the construct with mutated conserved 9-bp TR5 elements. The 5′-GCCATCTTG-3′ element was mutated into 5′-GAACGCTTG-3′. **g** Homologs of *trans* factors that bind to TIRs with high match score by JASPAR prediction. **h** The luciferase reporter assay indicated that hsYY1 had the most significant transcriptional suppression of pdTIR among the high scored *trans* factors predicted by JASPAR. **i** Mini core *cis*-acting elements for TIR transcriptional activity, probes generated for EMSA and TIR elements for TIR-containing plasmids are indicated. **j** Representative gels of EMSA indicated that hsYY1 could bind to the YY1-binding motif-containing probes. The results are representatives from three independent experiments, with similar results. See also Supplementary Fig. 1F. The values of **b**–**e** and **h** are the means ± s.d. with $n = 3$ biologically independent experiments. A two-tailed, unpaired Student's *t*-test was used for the comparisons in **c**–**e** and **h**. *$P < 0.05$; **$P < 0.01$; ***$P < 0.001$; ****$P < 0.0001$. ns, not significant. For **b**–**e**, **h**, and **j**, source data are provided as Source Data file.

**YY1 binds to the *cis*-acting elements of *ProtoRAG*.** To find specific *trans*-factors that bind to the *cis*-acting elements of *ProtoRAG*, motif scanning using the core *cis* elements (TIR sequences on pGL-R1-6r and pGL-R2-5) as targets was performed using the *trans*-factor database JASPAR. Certain *trans* factors, such as Yin Yang 1 (YY1), Aryl Hydrocarbon Receptor Nuclear Translocator Like protein (ARNTL), BTB domain containing 33 (ZBTB33), Upstream Transcription Factor 1 (USF1), and Signal Transducer and Activator of Transcription 6 (STAT6), were identified with high scores for binding within/nearby the core *cis* elements (Supplementary Data 1). Homologs of these tested *trans* factors with conserved domain architectures were also found in the lancelet genome (Fig. 1g). To reveal their roles in the transcription of *bbRAG1L* and *bbRAG2L* genes, expression vectors of these *trans* factors together with the pdTIR construct were transfected into 293T cells, respectively. As shown in Fig. 1h, most of the tested *trans* factors had some roles in the transcription of the reporter of the pdTIR construct. However, among these *trans* factors, YY1 not only had the highest match score, but also significantly repressed transcription of the reporter most.

Human YY1 (hsYY1) is a *trans* factor with diverse and complex biological functions[23,24]. Abundant evidence has demonstrated that the conserved zinc-finger domain at the C-terminus of YY1 recognizes the consensus sequences of 5′-GCCATCTTG-3′ located in the genomes[25,26]. To further reveal the roles of YY1 in the transcription of *ProtoRAG*, we first analyzed the YY1-binding motif (5′-CGCCATNTT-3′) of both 5′ TIR and 3′TIR according to the *trans* factor database JASPAR. As shown, the YY1-binding motif is located near the mini core *cis* elements and includes the defined 9-bp conserved TR5 elements (5′-GCCATCTTG-3′) within the TIRs (Figs. 1i and 3a, and Supplementary Data 1). Thus, to verify the binding of YY1 to the TIRs of *ProtoRAG*, several probes with truncated TIR sequences were generated for electrophoretic mobility shift assays (EMSA; Fig. 1j). The results showed that nuclear extracts from 293T cells could bind to YY1-binding motif-containing probes (Fig. 1j and Supplementary Fig. 1F). When human YY1 (hsYY1) mAb was added to EMSA reactions, super-shift bands could be observed in electrophoresis gels (Fig. 1j), further supporting the interaction between hsYY1 and YY1-binding motif-containing probes.

**BbYY1 represses the transcription of *bbRAGL* genes.** To understand the role of lancelet YY1-like protein (bbYY1) in the activity of *ProtoRAG*, we then cloned the *bbYY1* gene by 5′ and 3′ RACE from *B. belcheri*. Sequence alignment showed that YY1 was highly conserved among species (Supplementary Fig. 2). Based on a comparison with hsYY1, we found that bbYY1 contained the conserved zinc-finger domain (ZNF), the REPO domain and an additional C-terminal domain (CTD) but lacked the transcriptional activation domain (Fig. 2a and Supplementary Fig. 2). Similar to hsYY1[27], bbYY1 was found to be a nuclear-located protein (Fig. 2b) and could bind to the YY1-binding motif within TIRs of *ProtoRAG* (Fig. 2c and Supplementary Fig. 3B, C). In contrast to *bbRAG1L* and *bbRAG2L*, which have extremely low abundances in lancelet[6], *bbYY1* showed a high abundance during the entire lifetime of the lancelet (Supplementary Fig. 3D, E).

To address whether bbYY1 could regulate the transcription of *bbRAG1L* and *bbRAG2L* genes, we co-transfected bbYY1 with pdTIR or pdTIR-Mu (with mutations in YY1 binding sites in TIR-FL sequence) constructs into 293T or 293T[shYY1] cells to test the transcription of the reporter gene. The results showed that the transcriptional activity of pdTIR was increased when hsYY1 was deficient, but suppressed when bbYY1 was overexpressed. Additionally, the deficiency of hsYY1 could be rescued by overexpression of bbYY1 or hsYY1 (Fig. 2e, f). Moreover, pdTIR-

Mu displayed a lower transcriptional activity when compared with pdTIR (Fig. 2d), further supporting that the YY1 binding motif within the core *cis*-acting element was important for the transcription of *bbRAG1L* and *bbRAG2L* genes.

To reveal how bbYY1 restricted the transcription of *bbRAG1L* and *bbRAG2L*, several bbYY1 truncated constructs were created and co-transfected with pdTIR construct into 293T cells. The results showed that the conserved REPO and ZNF domains on bbYY1 were important for its transcriptional inhibition of reporter genes (Fig. 2a, g). EMSA further confirmed that the ZNF domain of bbYY1 was essential for its binding to TIR-containing probes (Fig. 2c). Moreover, when the zinc-finger domain of bbYY1 was deleted (bbYY1_ΔZNF), the mutated bbYY1 was distributed widely in both the cytoplasm and nucleus (Fig. 2b and Supplementary Fig. 3A), suggesting the presence of a nucleus location signal in the ZNF domain of bbYY1. Thus, the distribution of bbYY1 in the nucleus and the binding of bbYY1 to the *cis*-acting element of *ProtoRAG* through its ZNF domain were important for the transcriptional repression of *bbRAGL* genes.

**BbYY1 effects on TIR-dependent DNA recombination ex vivo.** Excluding the regulation of gene expression, TIR of the transposon element plays different important roles in its function and biological behavior, as is the case for the TIRs Ac/Ds in maize, which can lead to wide genome rearrangements[28,29]. Excisions and insertions of TIR elements in rice can introduce gene copy number variations[30]. Previously, we found that the exact TIR sequences are important for the transposon excision of *ProtoRAG* and the subsequent host DNA rejoins (HDJs)[8]. Since bbYY1 can bind to the highly conserved TR5 element on TIR of *ProtoRAG*, we further considered whether the reaction between the TR5 element and bbYY1 might affect functions of *ProtoRAG* other transcriptional repression. To answer this question, we first generated several constructs containing a poly(A) sequence flanked by a pair of minimal TIRs or mutated minimal TIRs (described simply as 5′ and 3′ TIR below) to test the roles of TR5 in *ProtoRAG*-mediated recombination (Fig. 3a–c). The pTIRG8-substrate contained a pair of 5′ and 3′ TIRs, while the pTIRG10 was a mutated substrate with a deletion of TR5 in both the 5′ and 3′ TIRs. The pCptG construct contained a mutated TR5 element in the 3′ TIR. When bbRAG1L/2L were co-expressed with these substrates in 293T cells, the poly(A) transcription stop sequence could be excised and the host DNA could be resealed by non-homologous end joining (NHEJ)-related factors, leading to the expression of the green fluorescent protein (GFP) reporter (Fig. 3b, c). As the results show, the ratio of GFP-positive cells was significantly reduced when the TR5 element of *ProtoRAG* TIR was deleted or mutated (Fig. 3d and Supplementary Fig. 4A). Moreover, mutation of the conserved nucleotides in the YY1-binding motif led to a greater decrease in the recombination efficiency, indicating a positive correlation between the *ProtoRAG*-mediated recombination and the YY1-binding motif (Supplementary Fig. 4C, D). Since bbYY1 binds to *ProtoRAG* TR5, we then performed flow cytometry assays using the pTIRG8 construct in 293 T and 293T[shYY1] cells to examine whether bbYY1 could affect *ProtoRAG*-mediated recombination. The results showed that HDJs mediated by bbRAG1L/2L were decreased in 293T[shYY1] cells, and this deficiency could be rescued by the overexpressed bbYY1 (Fig. 3e and Supplementary Fig. 4B).

To further reveal how bbYY1 affected the *ProtoRAG*-mediated recombination, a bacterial colony assay using the pTIR104 substrate was performed. When the poly(A) terminal flanked by a pair of minimal TIRs was removed by bbRAG1L/2L, the chloramphenicol-resistant gene *CAT* (chloramphenicol acetyltransferase) on the construct pTIR104 could be expressed (Fig. 3c). We first transfected

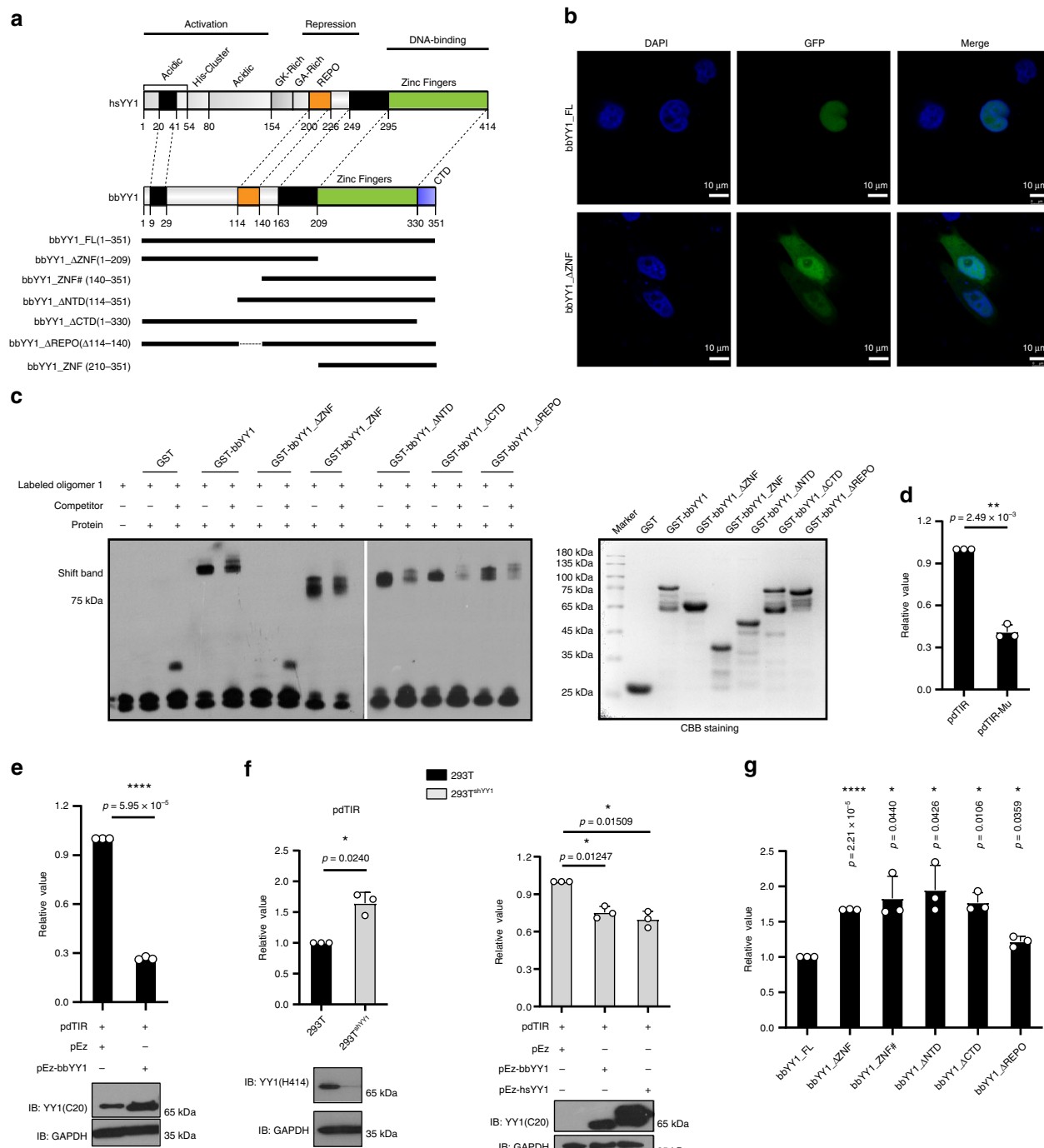

**Fig. 2 BbYY1 is involved in *ProtoRAG* transcription regulation. a** Domain architecture of hsYY1 and bbYY1 truncation proteins. **b** Laser confocal images of bbYY1 and truncates suggested a nuclear location signal in the zinc-finger domain (ZNF) of bbYY1. Images display the GFP-tagged bbYY1 and ZNF deleted truncates (bbYY1_ΔZNF) in HeLa cells. The displayed image represents two independent experiments with similar results. See also Supplementary Fig. 3A. **c** Representative EMSAs showed that the ZNF domain was necessary for the binding of bbYY1 truncations to the TIR probe. Left, EMSA gels. Right, SDS-PAGE of GST-tagged bbYY1 truncation proteins. The results are one representative from three independent experiments. **d** The luciferase reporter assay showed that the mutation of TR5 element on pdTIR decreased the transcription activity. pdTIR-Mu is a construct with a TR5 mutation, as shown in Fig. 1f. **e** BbYY1 suppresses the transcriptional activity of pdTIR in 293T cells by luciferase reporter assays. Upper, quantification of transcriptional activities of pdTIR with the expression of bbYY1 in 293T cells. Lower, western blotting of YY1 transfectants. YY1(C20) is a polyclonal antibody that can target both hsYY1 and bbYY1. **f** Luciferase assays showed that the transcriptional activity of pdTIR was improved with knockdown of hsYY1 in 293T cells, and reduced with overexpression of bbYY1 or hsYY1. IB, immunoblotting. YY1(H414) is a polyclonal antibody targeted to hsYY1. **g** Luciferase reporter assays showed that each functional domain of bbYY1 was indispensable for the transcriptional inhibition of TIRs. Transcriptional activities were measured by expression quantification of luciferase by transfection of pdTIR with different bbYY1 truncations into 293T cells. The values of **d–g** are the means ± s.d., with $n = 3$ biologically independent experiments. A two-tailed, unpaired Student's *t* test was used for comparisons. *$P < 0.05$; **$P < 0.01$; ***$P < 0.001$; ****$P < 0.0001$. ns, not significant. For **c–g**, source data are provided as Source Data file.

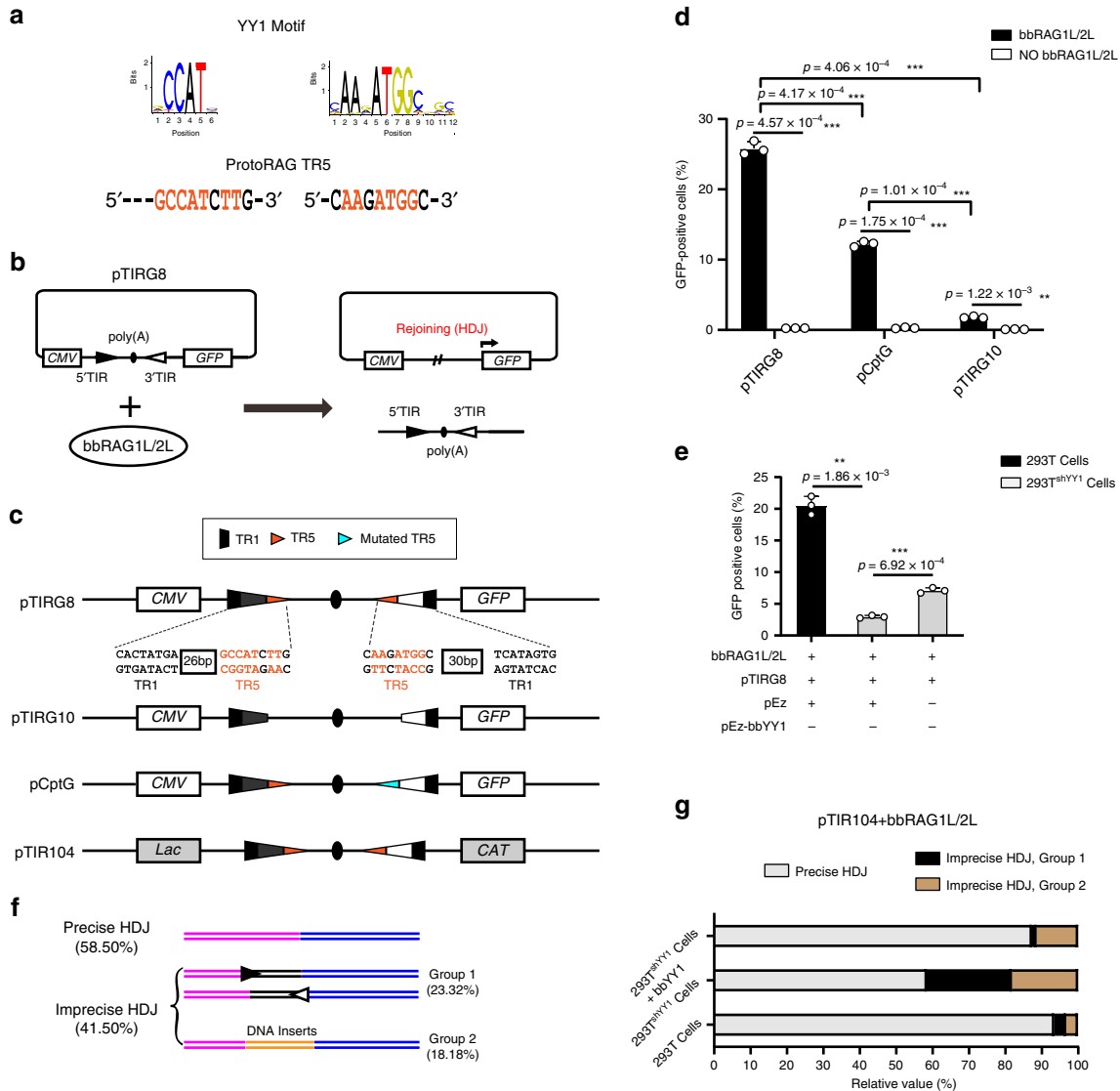

**Fig. 3 BbYY1 effects on precise TIR-dependent DNA recombination mediated by bbRAG1L/2L ex vivo. a** The TR5 element on *ProtoRAG* TIR is overlapped by a YY1-binding motif. The rainbow indicates the YY1-binding motif in the JASPAR database. Orange indicates critical base pairs in the YY1-binding motif. **b** Schematic diagram of the detection of TIR-dependent DNA recombination through FACS. Briefly, the pTIRG8 construct and indicated bbRAG1L/bbRAG2L expression vectors were co-transfected into 293T or 293T$^{shYY1}$ cells. After recombination, the polyadenylation poly(A) terminal signal would be excised off and the green fluorescent protein (GFP) would be expressed. The GFP signal could be detected by FACS. **c** Schematic diagram of TIR elements containing mutation plasmids pTIRG8, pTIRG10, pCptG, and pTIR104. The pTIRG8 and pTIR104 are WT TIR-containing constructs. The pTIRG10 is a substrate carrying deletion of both TR5 elements on TIRs. The pCptG is a construct with the mutated TR5 element on the 3′ TIR. *CAT*, chloramphenicol acetyltransferase gene; *CMV*, CMV promoter; *Lac*, Lac promoter. **d** Quantification of GFP-positive cells by flow cytometry showed that the TR5 element of *ProtoRAG* was important for effective TIR-dependent recombination ex vivo. pTIRG8, pTIRG10, and pCptG are as indicated in **c**. See also Supplementary Fig. 4A. **e** The flow cytometry assay showed that bbYY1 participated in TIR-dependent recombination. Quantification of GFP-positive cells by flow cytometry after transfection of pTIRG8 substrate and bbRAG1L/2L expression vectors into 293T cells (or 293T$^{shYY1}$ cells) with or without bbYY1 expression constructs. See also Supplementary Fig. 4B. **f** Diagrammatic sketch of precise and imprecise HDJ products in 293T$^{shYY1}$ cells. See also Supplementary Fig. 5A. **g** Percentage of precise and imprecise HDJ recombination in 293T and 293T$^{shYY1}$ cells. The values of **d** and **e** are the means ± s.d. with $n = 3$ biologically independent experiments. A two-tailed, unpaired Student's $t$ test was used for the comparisons. $*P < 0.05$; $**P < 0.01$; $***P < 0.001$; $****P < 0.0001$. ns, not significant. For **d**–**g**, source data are provided as Source Data file.

pTIR104 together with the expression constructs of bbRAG1L and bbRAG2L into 293T cells. Then, the recombined pTIR104 was recovered and transformed into *E. coli. DH5α*. After growing on the chloramphenicol-containing culture dish, a total of 153 or 253 random HDJ-containing plasmids from 293T WT or 293T$^{shYY1}$ cells after transposon excision were recovered and sequenced, respectively. The sequencing results showed that 93.46% of the clones recovered from WT 293T cells contained precise HDJs. However, only 58.50% of clones recovered from 293T$^{shYY1}$ cells

contained precise HDJs (Fig. 3f, g). As shown in Fig. 3f, precise HDJs indicated that both the 5′ TIR and 3′ TIR were completely excised from the pTIR104 construct. Imprecise HDJs were defined as HDJ products containing partial TIRs on the pTIR104 construct (group 1) and products with DNA inserts or DNA self-transposition after host DNA rejoining (group 2; also shown in Supplementary Fig. 5A). As shown in Fig. 3g, when bbYY1 was overexpressed in 293T$^{shYY1}$ cells, the ratio of precision HDJs was increased from 58.50% to 87.34%. Notably, the imprecise HDJs with

partial TIRs (imprecise HDJs, group 1) almost disappeared when the hsYY1 deficiency was recovered by bbYY1. PCR assays using the pdTIR construct containing a poly(A) terminus flanked by a pair of full-length TIRs (TIR-FLs) further confirmed the imprecise HDJs in hsYY1 knockdown 293T cells when compared with WT 293T cells (Supplementary Fig. 5B–D). Thus, bbYY1 benefited the precise TIR-dependent DNA recombination mediated by bbRAG1L/2L ex vivo.

**BbYY1 influences the TIR-dependent cleavage ex vivo.** Similar to V(D)J recombination, the TIR-dependent DNA recombination mediated by bbRAGL1/2L should include two major steps: DNA breaking and subsequent host DNA rejoining via the NHEJ repair mechanism[31]. The recovery of group 1 imprecise HDJs when bbYY1 was overexpressed in 293TshYY1 cells indicated that bbYY1 might affect the TIR recognition and cleavage steps. Thus, to further investigate how YY1 affected precise HDJ formation, we detected the cleavage products in TIR-dependent DNA recombination in WT 293T and 293TshYY1 cells. The ligation-mediated polymerase chain reaction (LM-PCR) method was used to detect the cleavages of TIR ends (TIREs), as shown in Fig. 4a. If the cleavage occurred at the borders of both 5′ and 3′ TIRs in 293T or 293TshYY1 cells, the intact TIR pair would be completely removed from pTIR104. When the recombined constructs were harvested as PCR templates, the right TIRE could be amplified by LM-PCR. In contrast, if the cleavage occurred inside the TIRs, the group 1 imprecise HDJs could be observed, and the short TIRE could be amplified. As shown in Fig. 4b, c, more TIREs could be detected in WT 293T than 293TshYY1 cells by LM-PCR detection, which is in line with our flow cytometry data shown in Fig. 3e. Moreover, consistent with our previous observation that bbRAG1L/2L prefers a single cleavage at the 3′ TIR in vitro[6], more 3′ TIREs than 5′ TIREs were obtained from the LM-PCR analysis (Fig. 4b, c). Additionally, short 5′ TIREs could be detected in 293TshYY1 samples from LM-PCR (Fig. 4b, c). Further sequencing analysis also supported that YY1 deficiency in 293T cells could result in short 5′ TIREs (Fig. 4d). In brief, these observations suggest that YY1 can influence the cleavage step in TIR-dependent DNA recombination mediated by bbRAG1L/2L.

**BbYY1 helps bbRAGL to recognize TIR specifically.** Since the cleavage step could be further divided into target DNA recognition, nicking, and hairpin formation[32], to further explore the influence of bbYY1 on bbRAGL-mediated TIR recognition ex vivo, a pSel G-mCh construct was designed and co-transfected with bbRAG1L/2L expression vectors into WT 293T or 293TshYY1 cells. As Fig. 4e illustrates, the GFP would be expressed when transposon excision occurred at the borders of 5′ TIR and the mutated 3′ TIR of TR5, while mCherry would be expressed when transposon excision occurred by recognition of the WT TIR pair. Thus, the ratios of GFP or mCherry-positive cells would indicate the recognition features of bbRAG1L/2 L for distinct TIRs through the fluorescence-activated cell sorting technique (FACS). The results showed that the ratio of mCherry-positive cells was decreased to ~1/6, and the ratio of GFP-positive cells was stable when hsYY1 was knocked down in 293T cells. Moreover, both ratios of GFP and mCherry-positive cells were increased when bbYY1 was overexpressed in 293TshYY1 cells. Moreover, the ratio of mCherry-positive cells increased to a greater extent than did that of GFP-positive cells when bbYY1 was overexpressed (Fig. 4f and Supplementary Fig. 4E), indicating that bbRAGL preferred to recognize right TIRs for recombination with assistance from bbYY1 ex vivo.

To further confirm this conclusion, we explored the recognition characteristics of bbRAGL to TIR-like DNA by DNA

pulldown assays[33]. As shown in Supplementary Fig. 6A, the complex of bbRAG1L/2L, but not bbRAG1L or bbRAG2L alone, could be pulled down by biotin-labeled 5′ or 3′ TIR. Moreover, the bbRAG1L/2L complex could also recognize 23-RSS, indicating that bbRAG1L/2L might bind to an unspecific DNA in vivo, which should benefit transposition targeting. Since bbYY1 binds to TR5 elements on TIRs of *ProtoRAG*, we next examined whether TR5 deficiency would affect the binding of bbRAGL to TIRs. The bbRAG1L/2L expression vectors and the bbYY1 expression vector were then co-transformed into 293T cells for DNA pulldown assays. The results showed that more bbRAG1L/2L and bbYY1 were pulled down by WT TIRs but not the TR5-mutated TIR (Bio-5′ TIR-Mu; Fig. 4g), indicating that bbYY1 helped bbRAGL recognize WT TIR specifically. Collectively, these results suggested that bbYY1 benefited the correct TIR targeting of bbRAG1L/2L in vivo.

**BbYY1 improves TIR-TIR joint formation.** As we have shown above, bbYY1 benefited HDJ formation by facilitating precise TIR targeting. Moreover, the improvement of HDJs may also result from efficient DNA repair[31]. Since the excised TIR-containing DNA can also be rejoined to form TIR-TIR joints (TTJs)[6], we then used a TIR reoriented construct based on pTIRG8 (named pTIRG8-ivt) to test whether bbYY1 could influence TTJ formation. We first co-transfected the pTIRG8-ivt vector with the *bbRAG1L/2L* and *bbYY1* expression vectors into 293 T or 293TshYY1 (Fig. 5a). Then, FACS assays were performed to show that the ratio of GFP-positive cells was greatly decreased when YY1 was knocked down in 293T cells. Such a reduction could be rescued to an almost equivalent level when bbYY1 was overexpressed in 293TshYY1 cells (Fig. 5b and Supplementary Fig. 4F), suggesting that bbYY1 promoted the formation of TTJ products ex vivo. Previously, the loss of hsYY1 or mouse YY1 has been demonstrated to affect the expression of many genes in the DNA repair pathway[34]. Some biochemistry evidence has indicated that YY1 improves the broken DNA repairing[35,36]. HsYY1 also participates in homologous recombination-based repair, chromatin remodeling, and intergenic connections by interacting with the INO80 complex and RNA-binding proteins, respectively[34,37–39]. Thus, the reduced HDJs and TTJs in 293TshYY1 cells might be due to its defect in DNA repair pathways.

**BbYY1 suppresses TIR-dependent transposition.** BbYY1 benefits the TTJs, which may result in the reduction of the *bona fide* transposition activity of *ProtoRAG*[6]. To confirm whether bbYY1 was also involved in TIR-dependent transposition, an ex vivo intermolecular transposition assay was performed. The donor plasmid pTet-dTIR containing the tetracycline resistance gene flanked by a pair of full-length TIRs (TIR-FLs) and the target plasmid pEGFP-N1 containing a kanamycin resistance gene were co-transfected along with bbRAG1L/2L expression vectors and the bbYY1 expression vector into 293T WT or 293TshYY1 cells. After transfection, DNA products were recovered from the cell lysate and transformed into bacteria to determine the transposition efficiency (Fig. 5c). Then, KanR/TetR colonies were selected for sequencing analysis. Compared with WT 293T cells, the bbRAG1L/2L-mediated transposition efficiency was increased ~2-fold in 293TshYY1 cells but suppressed when bbYY1 was expressed in 293TshYY1 cells (Fig. 5d). We also found that more 5-bp TSD occurred in 293 TshYY1 cells than WT 293T cells (Fig. 5e). However, the distribution of transposition sites and GC preference in TSD were similar in WT and YY1 knockdown cells (Fig. 5f and Supplementary Fig. 6B). These results indicated that bbYY1 could suppress TIR-dependent transposition without affecting the selection of transposition sites. Taken together, we

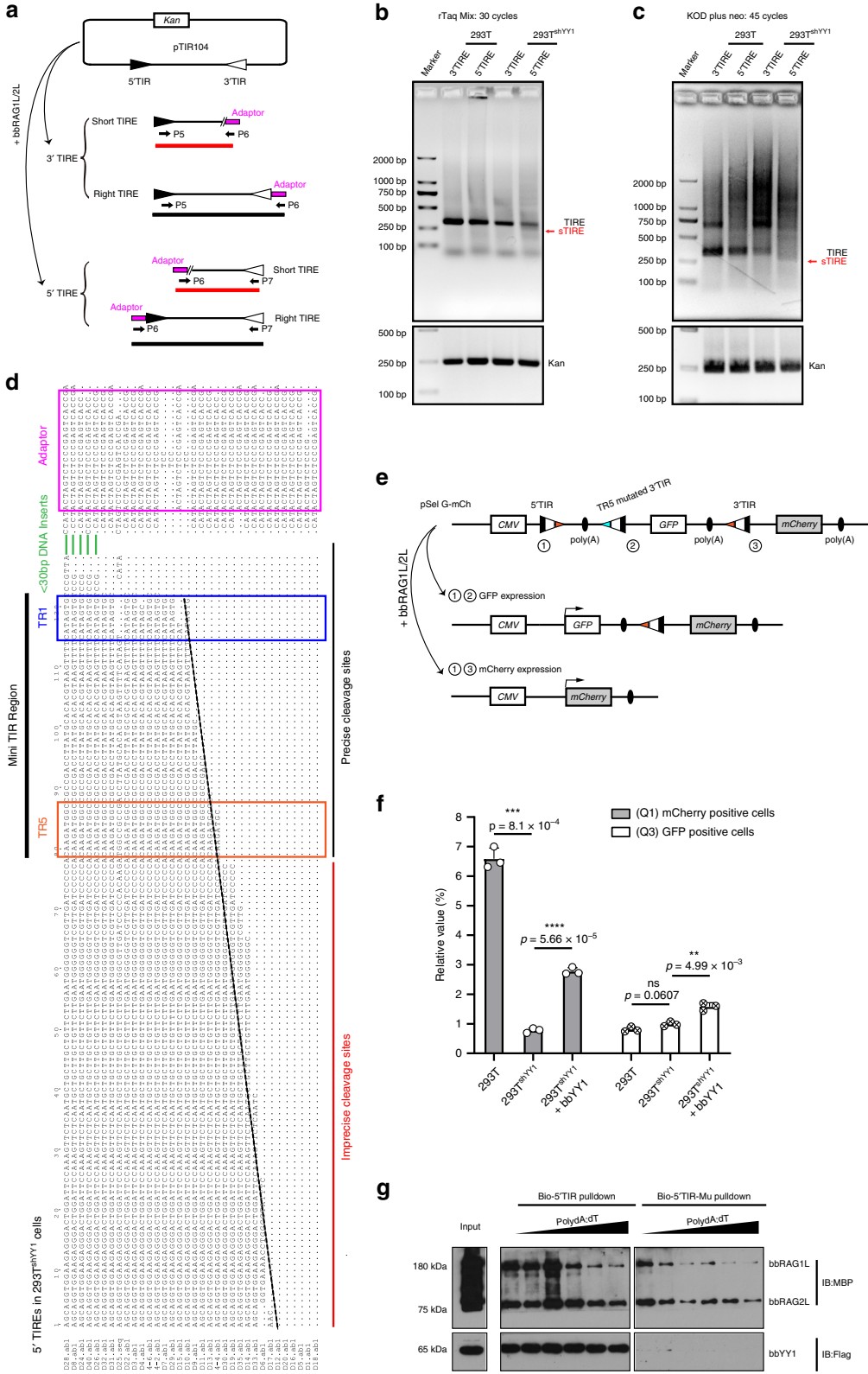

thought that the advantages of bbYY1 for TTJ formation perhaps contributed to such transposition suppression.

**BbYY1 interacts with bbRAG1L/2L.** Since bbYY1 benefited the accessible recognition of bbRAG1L/2L to correct TIR and suppressed the TIR-dependent transposition activity, we

investigated whether these regulations were accompanied by the interaction between bbYY1 and bbRAG1L/2L. Thus, co-immunoprecipitation (co-IP) and GST-pulldown assays were conducted and showed that bbYY1 indeed interacted with the bbRAG2L or bbRAG1L/2L complex through its zinc-finger domain (ZNF; Fig. 6a–e and Supplementary Fig. 6E, G). Furthermore, given that bbYY1 repressed *ProtoRAG* transcription,

**Fig. 4 BbYY1 contributes to precise TIR ends (TIREs) by aiding the recognition of appropriate TIRs. a** Illustration of the LM-PCR method for detecting cleavage sites of TIR substrate after bbRAG1L/2L-mediated recombination. Right TIRE happened when cleavages occurred at the borders of both 5′ and 3′ TIRs, while short TIRE happened when cleavages occurred between the 5′ and 3′ TIRs. P5, P6, and P7 indicate specific PCR primers. The adapter and these primers are listed in Supplementary Table 1. **b, c** The upper gels show that fewer and shorter TIREs occurred after TIR-dependent recombination in 293T$^{shYY1}$ cells. sTIRE, short TIR ends. The lower gels show the expression of the normalized control gene (Kanamycin resistance gene, indicated as *Kan*) on pTIR104. LM-PCRs were performed using rTaq DNA polymerase with 30 cycles (**b**) or KOD plus neo DNA polymerase with 45 cycles (**c**). The representative LM-PCR gels represent three independent duplications. For more details regarding the PCR procedures, see the Methods section. **d** Alignment of 5′ TIREs after TIR-dependent cleavage in 293T$^{shYY1}$ cells showed a broad range of imprecise cleavage, indicating the formation of group 1 imprecise HDJs in 293T$^{shYY1}$ cells. **e** Schematic diagram illustration of different TIR signal recognition mediated by bbRAG1L/2L in vivo. The recognition and cleavage occurring between sites 1 and 2 or between sites 1 and 3 would lead to GFP or mCherry expression, respectively. **f** Percentage of GFP- and mCherry-positive cells after bbRAG1L/2L-mediated recombination in 293T and 293T$^{shYY1}$ cells using the pSel G-mCh construct as **e** indicated. The values represent the means ± s.d., with $n = 3$ biologically independent experiments. A two-tailed, unpaired Student's *t* test was used for comparisons. *$P < 0.05$; **$P < 0.01$; ***$P < 0.001$; ****$P < 0.0001$. ns, not significant. See also Supplementary Fig. 4E. **g** The DNA pulldown assay showed that greater abundances of bbRAG1L/2 L and bbYY1 were pulled down by the WT 5′ TIR (bio-5′ TIR) but not the TR5-mutated 5′ TIR (bio-5′ TIR-Mu) in the presence of increasing poly dA:dT. The blots represent three independent duplications. For **b**, **f**, and **g**, source data are provided as Source Data file.

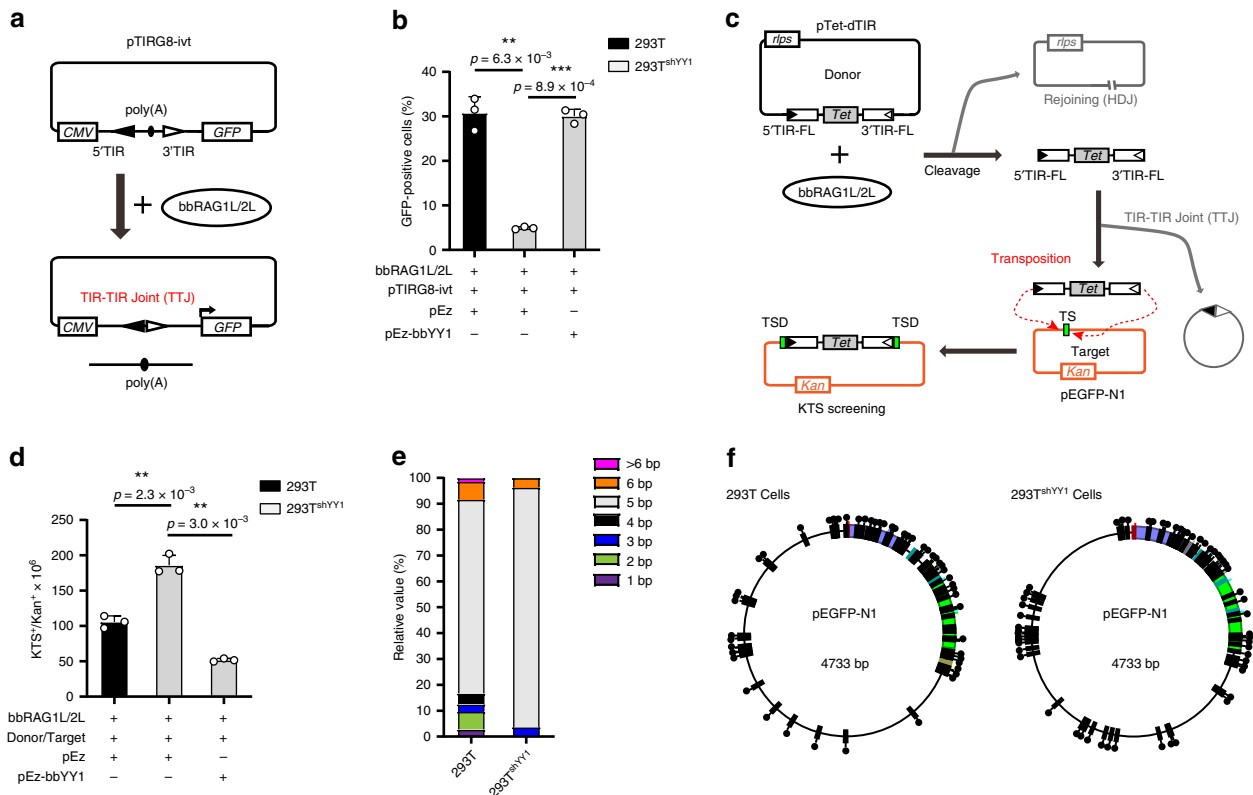

**Fig. 5 BbYY1 improves TTJ formation and suppresses TIR-dependent transposition. a** Schematic of the TTJ detection assay. Quantities of GFP-positive cells were used to indicate TTJ formation by transfecting a pair of reversed TIRs containing substrate pTIRG8-ivt with bbRAG1L/2L expression vectors into 293T or 293T$^{shYY1}$ cells. **b** The flow cytometry assay showed that bbYY1 promoted TTJ formation based on an analysis of the quantification of GFP-positive cells in 293T and 293T$^{shYY1}$ cells. See also Supplementary Fig. 4F. **c** Schematic diagram of intermolecular transposition. A donor plasmid (pTet-dTIR) containing the tetracycline resistance gene (*Tet*) flanked by the full-length TIR (TIR-FL) pairs and a target plasmid (pEGFP-N1) containing the kanamycin resistance gene were co-transfected with bbRAG1L/2L expression vectors into 293T or 293T$^{shYY1}$ cells. When bbRAG1L/2L-mediated DNA cleavage and transposition occurred, the *Tet* was removed from the donor plasmid (pTet-dTIR) and transposed into the target plasmid pEGFP-N1. The transposition efficiency could be quantified by survival rates of colonies on Kanamycin-Tetracycline-Streptomycin (KTS) resistance plates after bacterial transformation. Streptomycin *(str)* resistance was reduced by the *rpsL* gene in donor plasmid. TS, target site; TSD, target site duplication; *Tet*, tetracycline resistance gene. **d** Ex vivo intermolecular transposition assays showed that bbYY1 could suppress transposon efficiency according to the quantification analysis of transposition frequency of indicated plasmids in 293T and 293T$^{shYY1}$ cells. For more details regarding intermolecular transposition, also see the Methods section. **e** TSD types of intermolecular transposition products occurred in 293T and 293T$^{shYY1}$ cells. **f** The distribution of transposition sites generated by plasmid-to-plasmid transposition mediated by bbRAG1L/2L. The values shown in **b** and **d** are the means ± s.d., with $n = 3$ biologically independent experiments. A two-tailed, unpaired Student's *t* test was used for comparisons. *$P < 0.05$; **$P < 0.01$; ***$P < 0.001$; ****$P < 0.0001$. ns, not significant. For **b** and **d**–**f**, source data are provided as Source Data file.

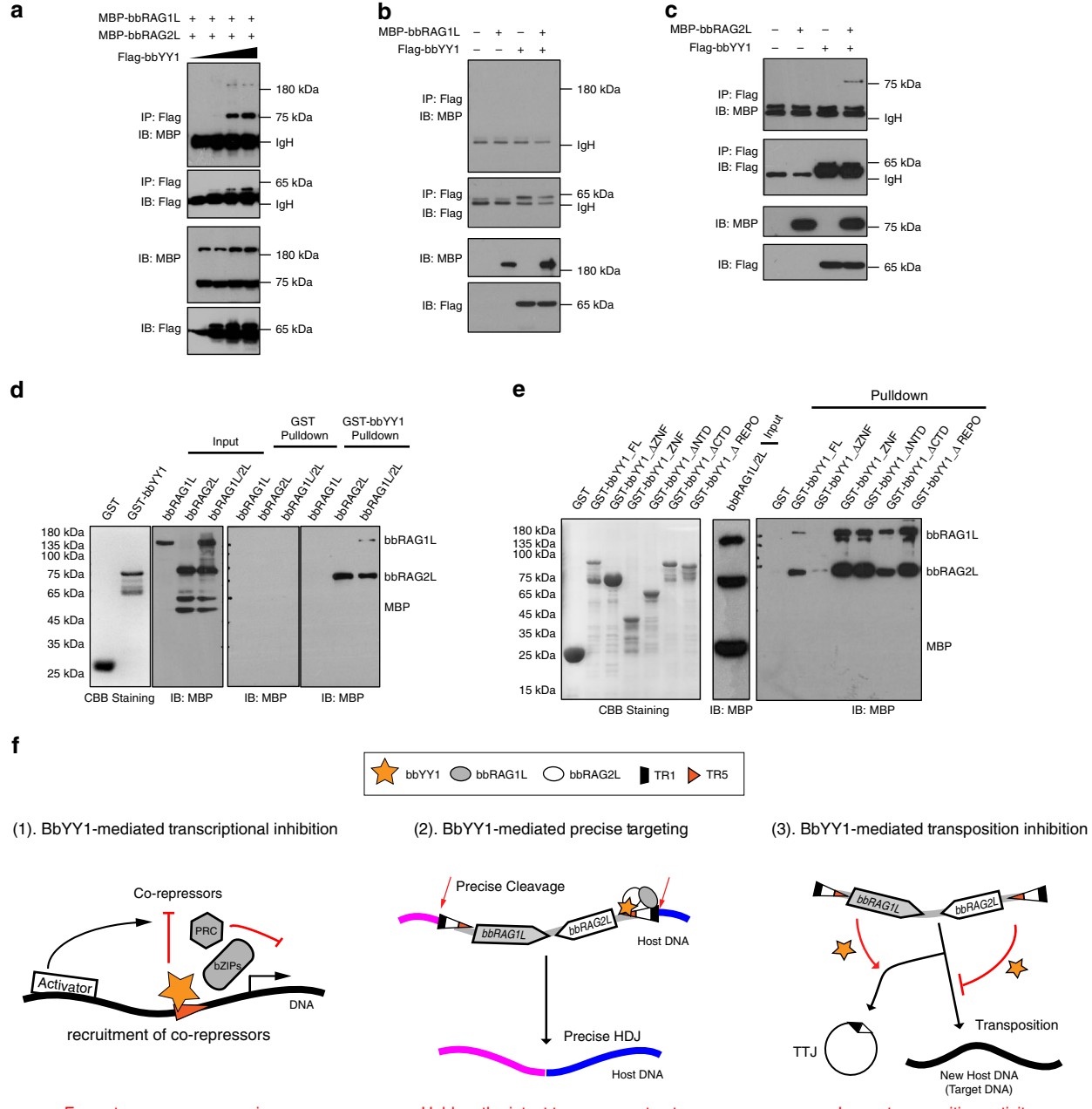

**Fig. 6 BbYY1 interacts with bbRAG1L/2L. a** The co-immunoprecipitation (co-IP) assay indicated that bbYY1 interacted with bbRAG1L/2L in a dose-dependent manner. **b** The co-IP assay indicated that bbYY1 did not interact with bbRAG1L by transfecting MBP-tagged bbRAG1L and Flag-tagged bbYY1 expression vectors into 293T cells. **c** The co-IP assay revealed the interaction between bbYY1 and bbRAG2L by transfection of MBP-tagged bbRAG2L and Flag-tagged bbYY1 expression vectors into 293T cells. **d** The GST-pulldown assay indicated the direct interaction between bbYY1 and bbRAG2L (or bbRAG1L/2L complex). **e** The zinc-finger domain (ZNF) of bbYY1 is indispensable for the bbYY1-bbRAG1L/2L interaction. SDS-PAGE and western blotting of the indicated GST-pulldown assays. See also Supplementary Fig. 6E, G. The co-IP and GST-pulldown assays in **a–e** represent three independent duplications. **f** A summary diagram illustrates multiple roles of bbYY1 in regulating bbRAG1L and bbRAG2L expression and *ProtoRAG*-mediated DNA recombination and transposition. First, through recognition of the TR5 element on *ProtoRAG*, bbYY1 may recruit co-repressors or compete with activators to suppress *ProtoRAG* transposon expression, contributing to lower transposition activity and a reduced transposition footprint in the lancelet genome. Second, by interacting with bbRAG1L/2L, bbYY1 can mediate bbRAG1L/2L precise targeting to the correct TIR signal sequences, which would lead to precise cleavages and HDJs. Hence, bbYY1 probably protects the intact transposon structure of *ProtoRAG* against decay. Third, bbYY1 can attenuate TIR-dependent transposition but improves TTJ formation, leading to suppressed transposition activity of *ProtoRAG*. For **a–e**, source data are provided as Source Data file.

we then redesigned and performed luciferase reporter assays to confirm the interactions between bbYY1 and bbRAG1L/2L ex vivo, as shown in Supplementary Fig. 6C. The results showed that bbRAG2L or bbRAG1L/2L could attenuate the inhibition of

bbYY1 on luciferase transcription, in agreement with our co-IP and GST-pulldown results.

To further explore the interface of the bbRAG2L-bbYY1 interaction, the constructs containing mutations in the core ZNF

(cZNF) and replacement of certain charged residues on bbRAG2L were used to perform GST-pulldown assays (Supplementary Fig. 6D–G). The GST-pulldown results showed that the interaction between cZNF and bbRAG2L was greatly disrupted when the negative amino acids were mutated to positive lysines on coils between α-helixes and β-sheets of cZNF (Supplementary Fig. 6D, E). Moreover, a change in positive spots at the edges or flanking the bottom sides of bbRAG2L would also eradicate cZNF-bbRAGL interactions (Supplementary Fig. 6F, G). Thus, we considered that potential effects produced by charged amino acids among bbYY1 and bbRAG2L likely co-contributed to the interaction interfaces of bbYY1 and bbRAG1L/2L in association with TIR DNA.

## Discussion

The discovery of *ProtoRAG* suggested that it is still alive but with extremely weak activity in the lancelet genome[6]. Here we further showed that *ProtoRAG* is a TIR-dependent and self-activated DNA transposon. Interestingly, we identified lancelet YY1 as a *trans* factor that binds to the core *cis*-acting elements of *ProtoRAG* to finetune its function and activity in the following aspects (Fig. 6f).

First, we found that the transcription of *bbRAGL* could be inhibited by lancelet YY1. Considerable evidence has demonstrated that mammalian YY1 can repress transcription of target genes by recruiting co-repressors or replacing activators or interfering with adjacent activators on promoters of target genes[40]. By analyzing the functional domains of bbYY1, we found that it lacked the transcriptional activation domain but contained the conserved REPO domain and zinc fingers. Furthermore, we found several highly conserved co-repressors in the lancelet genome, such as homologs of polycomb repressive complex (PRC) and cAMP-responsive element binding protein/activating transcription factor (CREB/ATF; Supplementary Fig. 7A, B), which have been demonstrated to form complexes with YY1 to repress the transcription of target genes[41,42]. These analyses indicated that lancelet YY1 might mediate transcription inhibition of *bbRAG1L* and *bbRAG2L* via its typic repression domain REPO and zinc fingers by recruiting PRC2 homologs or bZIP-containing co-repressors like its mammalian counterparts (Fig. 6f). Certain evidences have demonstrated that zinc-finger proteins[43], such as zinc-finger protein 91 (ZNF91) and ZNF93, can lead to transcriptional silencing of retrotransposons by recruiting repressive histone modifications[44]. Since YY1 is a zinc-finger containing protein, it may also repress the transcription of *bbRAGL* similar to that of ZNF91 or ZNF93. Alternatively, lancelet YY1 may also compete with an unidentified transcriptional activator to suppress the transcription of bbRAGLs since the deletion of the YY1-binding motif on TIR also reduced the transcription of bbRAGL genes (Fig. 2d). In addition to YY1, many other *trans* factors may also regulate the transcription of *ProtoRAG*, such as USF1 and ZBTB33 (Fig. 1h), indicating a complicated network of *ProtoRAG* regulation, which requires further study in the future.

Second, in mammalian cells, transposition mediated by mammalian RAG could be suppressed by the plant homeodomain (PHD) finger in the C-terminal portion of RAG2. PHD prevents RAG-mediated transposition by blocking the capture of unrelated target DNA when coding DNA is present, according to in vitro evidence[45]. Moreover, the GTP concentration, target site selectivity, and conformational change in RAG proteins also suppress RAG-mediated transposition in vivo[46–48]. Since *Proto-RAG*-mediated transpositions are deleterious to the host genome, *ProtoRAG*-mediated transposition should be highly restricted. As we have reported in our previous research, lancelet RAG2L lacks

the C-terminal PHD domain[6]. Thus, other mechanisms should have restricted the in vivo transposition activity of *ProtoRAG*. Recently, directed evolution of key amino acids in RAG transposases, leading to the host domestication of RAG from transposase to recombinase during evolution, has been confirmed[7]. Here, we further proposed another important regulatory mechanism for confining the transposition activity of *ProtoRAG* dependent on bbYY1. As shown in Fig. 5b, lancelet YY1 could promote TTJ formation, which is an important restriction of *ProtoRAG*-mediated transposition[6]. Additionally, mouse RAG2 has been found to enforce T-form DNA distortion and attenuate unwanted DNA transposition probably through the six-bladed Kelch structure[49]. Since lancelet RAG2L also has a similar six-bladed Kelch structure[7], the interaction between bbYY1 and bbRAG2L may influence the dynamic conformation and disintegration of the DNA-strand transfer complex, which would lead to the inhibition of *ProtoRAG*-mediated transposition. Moreover, many lines of evidence have shown that the host restrains transposition activity of transposons by repressing its transposon expression. For instance, Hfq can repress the expression of the Tn5 transposon[13], while LINE-1 is restricted by methylation of its CpG islands[50]. The transcriptional suppression of *bbRAG1L* and *bbRAG2L* by the ubiquitously expressed bbYY1 might cause a constitutive repression of the transposition of *ProtoRAG* from embryos to adults in lancelets, resulting in the maintenance of lancelet genome stability.

Third, in addition to its effect on repressing the transcription and transposition of *ProtoRAG*, we found that bbYY1 prevented imprecise TIR cleavages by helping the lancelet RAG1L/2L complex to target correct TIRs. As in mouse RAG, the nonamer binding domain (NBD) of RAG1 is responsible for the recognition of the nonamer on RSS specifically[51,52]. However, the NBD homologous domain of bbRAG1L (named NBD*) is highly divergent from that of mouse RAG1[6], and the conserved 9-bp TR5 element on *ProtoRAG* TIR is specific to lancelet (Supplementary Fig. 1G). We thus proposed that lancelet YY1 might play a role similar to that of mouse RAG1 NBD to help lancelet RAGL precisely target TIR sequences. Furthermore, as noted above, lancelet YY1 lacks the activation domain. Thus, the domain acquisition of vertebrate YY1 and RAG2 in evolution may have co-contributed to the evolution of the RAG system in vertebrates.

In conclusion, we propose that lancelet YY1 not only inhibits the transcription and transposition of *ProtoRAG* but also benefits HDJ and TTJ formation and helps the RAG1L/2L complex precisely target TIRs (Fig. 6f). The study here helps us to understand how *ProtoRAG* is regulated to maintain its long-time survival and genomic stability in the lancelet germline, which may shed new light on the coordination between TE regulation and genomic stability. Moreover, the functional regulation by bbYY1 or other *cis* and *trans* elements may help us to understand the host restriction or domestication of an active ancestral RAG transposon to host genes specific to the lymphatic system in vertebrates.

## Methods

**Animal, cells, antibodies, and related expression vectors**. Adult Chinese amphioxus (~1 year old) *B. belcheri* were captured by using dense nets from the sea area nearby Zhanjiang city, China. After capturing, the amphioxus were put into a sea water-containing tank and transported to the laboratory. During transportation, the tank was kept at 18–25 °C. The captured amphioxus were cultured in a laboratory incubator under modeled wild conditions. All the experimental protocols for handling of adult Chinese amphioxus were approved by the Institutional Animal Care and Use Committee of Sun Yat-sen University, Guangzhou, China. All relevant ethical rules regarding the animals were compliant in this study. The 293T and Hela cell lines from ATCC, were maintained in DMEM (Gibco) supplemented with 10% FBS at 37 °C under 5% $CO_2$. Transfections were performed using JetPRIME (cat.: 114–15, PolyPlus-transfection Bioparc.) according to the manufacturer's instructions. Antibody reagents were purchased from the indicated manufacturers and diluted to

appropriate concentrations for blotting analysis. Anti-Flag (1: 5,000; 66008-2-Ig, Proteintech), anti-GAPDH (1:10,000; 60004-1-Ig, Proteintech), anti-GST (1: 5000; 71097-3, Merck), anti-His (1: 5000; 70796-3, Merck), anti-MBP (1: 5,000; 66003-1-Ig, Proteintech or 1: 5,000; M 6295, Sigma), anti-YY1 (1: 5,000; H414; sc-1703, Santa Cruz), anti-YY1 (1: 5000; C20; sc-281, Santa Cruz), goat anti-mouse IgG-HRP (1:10,000; HA1006, HuaBio) and goat anti-rabbit IgG-HRP (1:10,000; HA1001, HuaBio) were used in this study. The pTT5-MBP-bbRAG1L and pTT5-MBP-bbRAG2L vectors[6,7] used for bbRAG1L/2L protein expression were gifts from Prof. David G. Schatz's lab. The *hsYY1* (cat.: EX-F0023-M12), *hsUSF1* (cat.: EX-F0238-M12), *hsARNTL* (cat.: EX-Z1347-M12), *hsZBTB33* (cat.: EX-T1082-M12), and *hsSTAT6* (cat.: EX-Y2605-M12) genes were cloned and constructed into the pEz-Flag (cat.:EX-NEG-M12) vectors for protein expression. These expression vectors were purchased from iGene Biotechnology Co., Ltd.

**Luciferase reporter assays.** In all, 293T cells were plated in 12- or 24-well plates, and after 24 h of culture, they were transfected with 500 ng DNA mixture per well. The mixed DNA contained indicated amounts of expression vectors and 30 ng phRL-TK plasmid (Promega). To normalize the transfection efficiency, deficiencies of expression plasmids were filled with empty plasmids. All samples were measured using the luciferase reporter assay system (Promega) according to the manufacturer's instructions. Values are represented as means of relative simulations for a representative experiment from three independent experiments ($n = 3$), each performed in triplicate wells.

**Electrophoretic mobility shift assays.** Biotin-labeled oligomer and unlabeled competitor oligomer were synthesized by Thermo Fisher. These oligomers were annealed to probes in Tris buffer (10 mM Tris (pH = 8.0), 1 mM EDTA, and 50 mM NaCl) by PCR incubation. Briefly, oligo nucleotides were incubated at 95 °C for 5 min and then gradually chilled down to room temperature. Probes were prepared and stored at a concentration of 1 pmol $\mu L^{-1}$ and diluted to 10 fmol $\mu L^{-1}$ immediately before use. The EMSA probe sequences are listed in Supplementary Table 1.

Nuclear proteins were extracted from the 293T cells according to the NE-PER® Nuclear extract protocol (cat.: 78835, Thermo Fisher). Volumes of 2–3 μL nuclear extracts or 500 ng–5 μg purified bbYY1 protein per 20 μL binding reaction were used. EMSA was performed according to the Light Shift Chemiluminescent EMSA Kit manual from Thermo Fisher (cat.: 20148). The binding reactions of protein and DNA were resolved on 8% non-denaturing polyacrylamide TBE gel and electrophoretically transferred to nylon membrane. The biotin-labeled DNA was finally detected using the chemiluminescence method.

**Characterization of potential TSSs of *bbRAG1L* and *bbRAG2L*.** Transfection of 293T cells was performed with plasmids containing *bbRAG1L* and *bbRAG2L* flanking sequences (pGL-R1-1, pGL-R2-1). At 24 h post-transfection, total RNA was isolated from these cells. Then, cDNA was synthesized from the total RNA using a PrimeScript RT-PCR kit (cat.: RR055B, Takara Bio.) according to the manual. 5′ RACE PCR was further conducted using well-designed primers for the identification of potential transcription start sites (TSSs). The RACE primers are listed in Supplementary Table 1.

**Characterization of bbYY1.** For *bbYY1* gene cloning, the total RNA was then isolated from adult lancelet tissues using TRIzol (cat.: 11667165001, Roche). In total, three male and three female adult amphioxus were randomly selected, quickly frozen, and euthanized by liquid nitrogen in one round of the experiment. cDNA was synthesized from the total RNA using a PrimeScript RT-PCR kit (cat.: RR055B, Takara Bio.) according to the manual. Specific primers targeting the *bbYY1* gene were designed and synthesized according to the prediction of the lancelet genome database. 5′ RACE and 3′ RACE PCR were then conducted for amplification of the *bbYY1* gene from the newly synthesized cDNA of lancelet. The RACE PCR products were recovered from the DNA gel and inserted into the pGEM-T-easy vector for further sequencing. The identified *bbYY1* gene fragments were finally assembled into a complete *bbYY1* gene. The *bbYY1* gene has been deposited in the NCBI under the accession numbers MF966513 [https://www.ncbi.nlm.nih.gov/nuccore/MF966513] and MF966514 [https://www.ncbi.nlm.nih.gov/nuccore/MF966514].

The *bbYY1* homologous genes were searched in the NCBI database using the BLASTN program. Certain homologs from representative species were then downloaded from the GenBank database for alignment using CLUSTALW[53]. The alignment results were finally colored using ESPript[54] for display.

For analysis of the *bbYY1* expression profile, qPCR was performed according to the ReverTra Ace qPCR RT Master Mix with gDNA Remover kit manual (cat.: FSQ-301, TOYOBO). The expression level of *bbYY1* was calculated using the comparative $2^{-\Delta\Delta Ct}$ method. Values are represented as means of three independent experiments, with each performed in triplicate reactions. The RACE primers and qPCR primers are listed in Supplementary Table 1.

For analysis of the bbYY1 cellular sublocation, expression vectors of GFP-tagged bbYY1 or bbYY1 truncations were transfected into HeLa cells. At 24 h after transfection, cells were fixed in a 4% formaldehyde solution, washed, stained for 5 min with DAPI, and then washed and imaged using a confocal microscope SP5 (LEICA).

**HsYY1 was knocked down by siRNA or shRNA in 293T cells.** Synthetic siRNA oligos targeting *hsYY1* and non-targeting control oligos were obtained from Guangzhou Ribo Bio Co. Ltd. The transfection of 293 T cells was performed with the indicated siRNA using Lipofectamine® RNAiMAX according to the manual (cat.: 11668-019, Thermo Fisher Scientific). Cells were harvested at 48 h post-transfection. The hsYY1 knockdown efficiency was analyzed by western blotting (using anti-YY1 (H414; sc-1703) or anti-YY1 (C20; sc-281) antibodies, Santa Cruz). Signals were normalized to GAPDH.

For 293T^shYY1 cell construction, three pairs of hsYY1 shRNA oligomers were synthesized and annealed according to the manual's instructions (Thermo Fisher Scientific). The lentiviral pLKO.1-shYY1 plasmids were first constructed. Viral particles were then prepared by transfecting psPAX2 and pMD2.G with pLKO.1-shYY1 plasmids into 293T cells. A more detailed description of the lentiviral package can be obtained from the instructions for Addgene Plasmid 10878. Finally, the constructed 293T^shYY1 cells were screened out via puromycin resistance and maintained as usual with puromycin (2 μg mL$^{-1}$) addition. The hsYY1 knockdown efficiency was confirmed by western blotting. Three pairs of hsYY1 shRNA oligomers are listed in Supplementary Table 1.

**Flow cytometry assays.** The 293T or 293T^shYY1 cells in 12 or 24-well plates were cultured and transfected with 100 ng substrate vectors and 200 ng protein expression vectors per well according to the requirements (including bbRAG1L/2 L protein expression vectors or the bbYY1 expression vector). After 48 h of culture, the cells were centrifuged, washed, and resuspended in PBS. The GFP signals were then analyzed using a Beckman CytoFLEX. For quantification of mCherry- and GFP-positive cells, the Beckman MoFlo Astrios EQs was used. CytExpert 1.2 and FlowJo 10 software were used for data analysis. All samples in each independent experiment were performed in triplicate. Values are represented as means of relative simulations for a representative experiment from three separate experiments ($n = 3$).

**Bacterial colony assays.** Bacterial colony assays with pTIR104 were conducted according to the following protocol. First, the pTIR104 (200 ng) vector together with the bbRAG1L (200 ng) and bbRAG2L (200 ng) expression vectors were co-transfected into 293T or 293T^shYY1 cells on a six-well culture plate. After the transfection, the cells were continually cultured for a further 48 h. Subsequently, the recombinant plasmid was recovered from the transfected cells through the alkaline lysis method and transformed into the *E. coli DH5α*. Kanamycin (100 μg mL$^{-1}$) and chloramphenicol (50 μg mL$^{-1}$) containing LB plates were used to select the positive clones. These positive clones were sent to sequencing for further analysis.

For the detection of HDJs from pdTIR, two iterations of PCR were performed. Briefly, after transfection of bbRAG1L/2L expression vectors, pdTIR together with hsYY1 siRNA were transfected into 293T cells for 48 h, and the recombinant pdTIR plasmids were isolated. The primer pair P1 and P2 (indicated in Supplementary Fig. 5C) was used to amplify HDJs after TIR-dependent recombination. The newly amplified HDJ products were recovered and inserted into pGEM-T-easy vector followed by their transformation into *E. coli. DH5α* cells. The positive bacterial colonies were then selected and subjected to a second PCR using primer pairs P3 and P4 (indicated in Supplementary Fig. 5C). This second series of PCR products were then resolved on 2% agarose gels for electrophoresis band analysis. Moreover, the selected positive clones were sent for further sequencing analysis. The PCR primers used are listed in Supplementary Table 1.

**Analysis of the 5′ and 3′ TIREs by LM-PCR and sequencing.** Briefly, bbRAG1L/2L expression vectors (2 μg) together with pTIR104 (2 μg) substrate were co-transfected into 293T or 293T^shYY1 cells according to the JETPRIME protocol (cat.: 114-15, polyplus-transfection Bioparc.). After 48 h of incubation, recombinant DNA was recovered from these cells using the alkaline-SDS lysis method (see the E. Z.N.A.® Plasmid DNA Mini Kit I protocol). For LM-PCR, adapters were prepared by annealing FM1 with FM2 on the PCR instrument; 200 ng of the recovered DNA was ligated to the well-prepared adapters and then used as a PCR template for the reactions. Primer pairs P5/P6 and P6/P7 were used for PCR amplification of the 3′ and 5′ TIREs, respectively (adapters and PCR primers are listed in Supplementary Table 1). The two different rTaq Mix (30 cycles, cat.: RR901A, Takara) and KOD-plus-neo (45 cycles, cat.: KOD-401, TOYOBO) DNA polymerases and corresponding PCR procedures according to the manuals were used for each sample test. The PCR products of TIREs were resolved by electrophoresis of 1.5–2% agarose gels. After gel electrophoresis, target bands (TIREs) were then recovered through gel extraction kits and subjected to TA cloning for further sequencing analysis.

**Analysis of DNA and protein sequences.** The TIRE sequencing data were filtered using SeqMan 7.1.0 (44.1) software (Gene Star). The HDJs were analyzed using ApE v2.0.49.10 software with manual adjustment of the alignment parameter settings. The CREB, ATF, and PRC protein sequences were downloaded from the public NCBI database and aligned against the Chinese lancelet genome database[55] [http://genome.bucm.edu.cn/lancelet/] for homolog searching via the BLASTP program with the default parameter settings. The highest scoring homologs from

the lancelet genome database were then downloaded. The online SMART server[56] [http://smart.embl-heidelberg.de/] was used for the prediction of the conservative domain architectures of these homologs. The MUSCLE in MEGA 5.2 or T-Coffee server[57] [http://www.tcoffee.org/] was used for evaluating and aligning DNA or protein sequences. Genedoc 2.7, BioEdit 7.0.5.2 software and ESPript 3.0 web server[54] [http://espript.ibcp.fr/ESPript/cgi-bin/ESPript.cgi] were used for the refinement or shading of the sequence alignments.

A co-crystal structure of the human YY1 zinc-finger domain bound to the adeno-associated virus P5 initiator was downloaded from the PDB database[58] (PDB: 1UBD [https://doi.org/10.2210/pdb1UBD/pdb]) and used for remodeling the core ZNF (cZNF) domain of bbYY1 (bbYY1_cZNF). The I-TASSER program[59] [https://zhanglab.ccmb.med.umich.edu/I-TASSER/] was used for the prediction of the bbYY1 structure. A model structure of the bbRAGL-3′ TIR synaptic complex with nicked DNA was downloaded from the PDB database[7] (PDB: 6B40 [https://doi.org/10.2210/pdb6B40/pdb]). BbRAG2L and bbYY1_cZNF were displayed using PyMOL (TM) 1.7.4.5 Edu software as required.

**Protein expression and purification.** For His-tagged bbYY1 expression and purification, the pET28a-bbYY1 plasmid was transformed into E. coli. BL21(DE3) bacteria. The transformed bacteria were cultured to an OD = 0.6–0.8, chilled to 18 °C and induced with 0.5 mM IPTG overnight. After induction, the bacterial cells were collected and sonicated for lysis. The cell lysis supernatant was purified through Ni-NTA resin (cat.: 30210, Qiagen) with binding buffer A (25 mM Tris (pH = 8.0), 500 mM NaCl, 40 mM imidazole, 1 mM PMSF, and 1 mM DTT). The resin was slowly washed with increasing imidazole concentrations to ~160 mM. The eluate was collected and concentrated in vials and dialyzed with dialysis buffer (25 mM Tris, pH = 7.5, 150 mM KCl, 10% glycerol, and 2 mM DTT) through ultra-filtration.

For GST-tagged bbYY1 (or GST) expression and purification, the pGEX-6p-1-bbYY1 (or pGEX-6p-1)-containing plasmid was transformed into E. coli. BL21 bacteria. After culturing and inducing (the conditions were similar to those used for His-tagged bbYY1), the cultured bacterial cells were homogenized with lysis buffer B (25 mM Tris, pH = 7.5, 1 M KCl, and 1 mM DTT). The supernatant of the lysate was then purified through Glutathione Sepharose 4B beads (cat.: 17-5113-01, GE). The beads were washed with 10 column volumes of lysis buffer B and then eluted to obtain protein using 10 mM GSH-containing buffer (25 mM Tris, pH 7.5, 0.5 M KCl, 1 mM DTT, and 10 mM GSH). After elution, the target protein was collected, concentrated, and dialyzed using the same conditions described above.

All purified proteins were divided into small aliquots and stored at −80 °C before use. Protein purity was determined by SDS-PAGE followed by Coomassie Brilliant Blue (CBB) staining. The protein concentration was determined using the Bradford method.

**Co-IP assays.** The 293T cells were plated on six-well dishes ($5 \times 10^6$ cells per well) and transfected with 3 µg DNA plasmid (1.5 µg for each expression vector; if the target expression vector was insufficient, parallel empty plasmid was added). At 48 h post-transfection, the cells were lysed using western and IP lysis buffer (cat.: P0013, Beyotime Biotechnology). The lysate was incubated with primary antibodies (0.5~1 µg Flag antibody (cat.:66008-2-Ig, Proteintech)) at 4 °C overnight and then incubated with Protein G Sepharose (cat.: 17061802, GE) for an additional 4 h at 4 °C. Analysis was conducted using SDS-PAGE electrophoresis followed by western blotting. The results were visualized using the ECL method (cat.: #C900376 and #C900377, Sangon Biotech (Shanghai) Co. Ltd).

**GST-pulldown assays.** Fragments of bbYY1 or hsYY1 were inserted into the pGEX-6P-1 vector for GST-fused protein constructions. The constructs were then transformed into E. coli. BL21 bacteria. The transformed bacteria were cultured at 37 °C to an OD = 0.6–0.8, chilled to 18 °C and induced with 0.5 mM IPTG. After 12 h of induction, the cells were collected, resuspended, and sonicated with lysis buffer (TBST with protease inhibitor cocktail). The lysate supernatant was incubated with 40 µL of Glutathione-Sepharose slurry (cat.: 17-5113-01, GE) for 1 h at 4 °C to pull down GST-fused protein.

The MBP-bbRAG1L/2L expression vectors were co-transfected into 293 T cells for MBP fusion protein preparation. The transfected cells were continually cultured 60 h after vector transfection and lysed using western and IP lysis buffer (cat.: P0013, Beyotime Biotechnology). The lysate supernatant was rotated with 40 µL of the above well-prepared slurry (containing GST or GST fusion protein) in TBST buffer (with protease inhibitor cocktail addition) at 4 °C overnight and washed four times with TBST. Finally, the slurry binding with proteins was eluted in Laemmli buffer (supplemented with 200 mM 2-mercaptoethanol) and resolved on 10% SDS-PAGE gels for further western blotting analysis. The ECL chemiluminescence method was used for the western blot analysis.

**DNA pulldown assays.** The method used was referenced from a previous description[33]. First, for protein preparation, the MBP-bbRAG1L/2L and Flag-tagged bbYY1 expression constructs were co-transfected into 293T cells for protein expression. After 48 h of culture, the associated proteins were extracted from 293T cell lysates using western and IP lysis buffer (cat.: P0013, Beyotime Biotechnology).

Second, for biotin-labeled probe preparation, the single-stranded biotinylated oligonucleotide and an equal quantity of antisense oligonucleotide were dissolved in deionized sterile water and annealed on the PCR instrument as described above for the EMSA method. Third, for DNA pulldown, 20 pmol biotin-labeled probes and appropriate competitor DNA (poly dA:dT or unlabeled 5′TIR) were added to 500 µL of the cell lysates. The ratios of competitor DNA and biotin-labeled probe were set to ranges of 1.25:1, 2.5:1, 5:1, 10:1, and 20:1. After mixing of the DNA and cell lysates, the mixtures were then set on a rotator for 30 min of pre-incubation at 4 °C. After pre-incubation, 30 µL streptavidin-agarose (cat.: S1638, Sigma or cat.: 20359, Thermo Scientific) was added to each mixture for an additional 1 h of incubation. Each sample was then washed with TBS (pH = 8.0, with protease inhibitor cocktails added (cat.: 5892791001, Roche)) by centrifugation at $1000 \times g$ for 2 min three times. The centrifuged agarose pellets were then collected, resuspended in 50 µL Laemmli sample buffer and resolved on 8% SDS-PAGE gels for western blotting analysis. Images were obtained using ECL chemiluminescence.

**Intermolecular transposition assays.** The ex vivo plasmid-to-plasmid transposition assay was performed as recently described[7]. As follows, each 4 µg pTT5-MBP-bbRAG1L and pTT5-MBP-bbRAG2L expression vector, together with 6 µg donor plasmid (pTet-dTIR), 10 µg target plasmid (pEGFP-N1), and 4 µg YY1 expression vector (or empty vector), was co-transfected into 293T or 293T$^{shYY1}$ cells using polyethyleneimine. The medium was changed after transfection of cells for 24 h. Cells were continuously cultured for another 36 h and collected. Plasmid DNA was extracted from cell lysates using the alkaline-SDS lysis method (see the E.Z.N.A.® Plasmid DNA Mini Kit II protocol). Each 300 ng DNA was then transformed into MC1061 bacterial cells, which were plated onto the kanamycin-tetracycline-streptomycin (KTS) or kanamycin plates. The transposition efficiency was determined by calculating the ratio of KTS resistant to kanamycin resistant clones. All the KTS resistant clones were sent for sequencing for further transposition site analysis.

**Statistical analysis.** The results are expressed as means (±s.d.). The unpaired Student's t test was used for comparisons. Values of $P < 0.05$ were considered statistically significant (two tailed), indicated by "*"; $P < 0.01$, indicated by "**"; $P < 0.001$, indicated by "***"; and $P < 0.0001$, indicated by "****". $P > 0.05$ was not significant and is indicated by "ns". All samples in each independent experiment were conducted in triplicate. Relative simulations for a representative experiment were from three separate experiments ($n = 3$) unless otherwise noted. The exact number of replicates and exact P values are indicated in figures or figure legends. Statistical analysis was performed using GraphPad Prism 8 and Microsoft Excel 2013.

**Reporting summary.** Further information on research design is available in the Nature Research Reporting Summary linked to this article.

## Data availability

The data that support the findings of this study are available within the article and Supplementary Information. The bbYY1 cDNA sequences have been deposited in the GenBank database under accession numbers MF966513 and MF966514. Partial sequences of BAC plasmid clone BAC73, which contains the complete coding sequences of the bbRAG1L and bbRAG2L genes, are available in DDBJ/ENA/GenBank under accession number KJ748699 with identifiers[6]. The 3D model structures of the human YY1 zinc-finger and bbRAGL which support this study are available from the PDB database under the accession codes: 1UBD and 6B40, respectively, with the identifiers[7,58]. Further data are available from the corresponding author upon request. Source data are provided with this paper.

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

## Acknowledgements

We thank Prof. David G. Schatz and Prof. Frederick W. Alt for helpful discussions. We thank Dr. Yuhang Zhang for suggestions on protein expression and purification. This work was supported by grant 2018SDKJ0302-2 from Marine S&T Fund of Shandong (S.Y.), grants 91231206 (A.X.), 81430099 (A.X.), 31970852 (S.Y.) and 31800740 (X.T.) from the National Natural Science Foundation of China, grant 2018YFD0900502 from Ministry of Science and Technology of the People's Republic of China (S.Y.) and grant 2017B030314021 from Guangdong Science and Technology Department (S.Y.).

## Author contributions

A.X., S.Y., and S.L. conceived of the study. S.L., S.Y., and A.X. designed the experiments. S.L. performed most of the experiments with assistance from X.G.; S.L., S.Y., and X.G. performed the data interpretation. S.Y. and A.X. advised on the experiments. X.T. contributed to making substrate constructs of pTIRG8-ivt, pTIRG8, pTIRG8m1-m9, and pTet-dTIR and provided suggestions for the flow cytometry experiments. W.Y. contributed plasmid pTIRG10 and assisted with the flow cytometry experiments. X.L. assisted with cloning of the *bbYY1* gene. S.Y. and S.L. drafted the manuscript. S.Y., S.L., S.C., and A.X. reviewed and edited the manuscript. All authors contributed to discussions.

## Competing interests

The authors declare no competing interests.
