## [Peer Review File · Nature Communications]

Reviewers' comments:

Reviewer #1 (VDJ, RAG transposition) (Remarks to the Author):

I liked this paper the first time around - it should be published.

The two recent Nature papers cited by the authors indicate the importance of the topic and the relevance of this work from the Dreyfuss lab. Publish after the minor corrections I suggested last time.

Reviewer #2 (AID, VDJ recombination) (Remarks to the Author):

These authors previously identified and characterized a ProtoRAG transposon in lancelet that is thought to represent an evolutionary precursor of the RAG recombinase system used by the adaptive immune system in jawed vertebrates. The ProtoRAG transposon is composed of genes encoding RAG1-like and RAG2-like proteins flanked by TIR sequences related to RSSs. Here they show that the TIR sequences contain promoter elements and binding sites for lancelet YY1, which represses ProtoRAG transcription, stimulates TIR-dependent excision and recombination, but suppresses TIR-dependent transposition in favor of TIR-TIR joining. YY1 functions in part by direct binding to bbRAG2L. They contend that YY1 regulation of ProtoRAG activity is beneficial for genome stability and was an early step in domestication of this transposon for use in the vertebrate immune system. This is, in principle, an intriguing and important contribution. However there are many problems with the presentation and some experimental shortcomings and inconsistencies that must be satisfactorily resolved before this manuscript should be given further consideration.

The manuscript is inexpertly written and in sufficiently poor English that it is sometimes very difficult to imagine exactly what the experiment is and what the authors are trying to say. They need to pay much greater attention to the presentation and work closely with native English speaking editors to improve the manuscript.

The title is a bit misleading, as there is nothing in this manuscript about lymphoid restriction. I would suggest either "...evolutionary beginning of the vertebrate RAG system" or "...evolutionary beginning of domestication of the vertebrate RAG system".

The discussion of RAG gene regulation in the 3rd paragraph of the Introduction (particularly 87-92) seems arbitrary and simplistic. With everything known about RAG gene regulation, why selectively call out NFATc1? What is the point? Why selectively call out the role of distal elements 5' of RAG2? Why, for the sake of their argument, is it relevant that the elements are asymmetrically disposed? In fact it is not clear from the writing what the authors mean by "asymmetrically cis genetic elements" if one did not know the literature already. Moreover, the obscure reference to region II will be understood by no one and will unnecessarily force readers to the literature.

The figure legends, in general, are incorrectly written. They often state the result of the experiment, but do not always state what was actually done in the experiment. For example, Fig 1 b,d,e should mention that these are luciferase experiments. See also Fig. 2d,f. Correct as necessary throughout. Fig 1b,d,e. No information is provided about replicates or statistical significance. Same is true for Fig S2D, S3D

Fig 1b. Luciferase orientation is depicted in Fig S1B but not here. Please be consistent.

Fig 1f. Please correct PolyA spelling.

Line 128. Both TSSs are said to be located within the TIRs, but Fig. 1G shows this to be true only for the 3' TIR. Please clarify.

Line 141-144. The authors argue that the TIR sequences drive transcription of the RAG1L and RAG2L. Although they have demonstrated promoter activity in luciferase assays, whether these promoters drive transcription of RAG1L and RAG2L is speculative. Can the authors demonstrate RAG1L and RAG2L transcripts initiating at these locations?

Fig 1h and all subsequent non-quantitative figures. The authors do not indicate whether the experiment was repeated with similar results.

Fig 2c. The fact that YY1 is inhibitory to transcription but mutation of the YY1 site causes reduced expression are, at face value, contradictory. The authors do not acknowledge this, leading to confusion. If the presumption is that YY1 competes with an unknown activator for this site, please state this hypothesis. Alternatively, YY1 could be an activator and the effect of YY1 knockdown could be an indirect effect.

Fig 2d. If this figure involves transfection of 293TshYY1 then the figure legend should say so. The authors should show levels of human YY1 protein in transfectants compared to 293T wild-type and knockdown.

Fig 2f. bbYY1 deletion mutants should be tested for binding to TIR DNA.

Fig 3b. I do not see where the authors describe precisely the mutations tested in the different constructs.

Line 259. "reduce shortness of TIR dependent excision"? Please explain. Imprecision in coding joints appears to be attributed to altered specificity of cleavage but it remains possible that this represents an NHEJ defect. This should be acknowledged.

Fig 4a, legend to Fig 4, "shorten" is inappropriate usage. Heading Fig 4b by "rTaq Mix: 35 cycles" and Fig 4c by "KOD plus neo: 45 cycles" is confusing because it is not mentioned in the figure legend. Are the names of the polymerases rTaq mix and KOD plus neo? The Methods indicate the former at 30 cycles but the figure says 35. Please explain. Use the figure legend to explain the experiment. If 4b and 4c are essentially repeats of the same experiment, the recovery of short TIRs is not reproducible. Has this experiment been repeated sufficiently to know if the results are consistent and that the quantitative differences are statistically significant? Is there a control PCR to normalize 293T to 293TshYY1? Conclusions about the influence of YY1 on cleavage need strengthening.

Fig S4GLines 291-3. By "aimed DNA recognizing" do you mean "target DNA recognition"? Since you have already concluded that YY1 promotes more abundant and more precise recombination and cleavage, what is the rationale for examining whether "YY1 deficiency would shift recognition sites of bbRAGL to TIRs". This does not seem to be sensible.

Lines 298-301. Everything about this sentence is incorrect: "However, unlike mouse RAG1/RAG2 complex, bbRAG1L/2L could recognize 23-RSS" – it should be: "like mouse RAG1/RAG2". and then "indicating that the binding of bbRAG1L/2L complex to 23-RSS DNA sequence is not species specific" – but this makes no sense since there is only one 23-RSS tested (mouse). There is no 23-RSS in lancelet. Please correct the first part of the sentence and there is no need to say anything further.

Figs S5G,H are uninterpretable. The text (304-305) describes a result assessing binding of bbRAG1L/2L to wild-type TIRs in presence of YY1. But the figure panels compare wild-type to mutant TIRs and there is no indication as to whether YY1 is transfected in all conditions or only in some. And the figure legend describes a different result that is never mentioned in the text, regarding a comparison of wild-type to mutant. Unfortunately the figure legend does not provide basic information on the nature of the experimental manipulation. Please synchronize the text, the figure and the figure legend so this becomes coherent. I can draw no conclusions.

Lines 318-321. Why is the ratio of GFP+ cells increased by YY1 if TR5 is mutated? In addition, it is stated that "the ratio of mCherry+ cells increased more than that of GFP+ cells when bbYY1 or hsYY1 was overexpressed" even though this does not appear to be true for the latter. Finally Fig 4e, although performed three times, does not provide any statistical assessment of significant differences.

Fig 5d. Please add CptG and TIRG10 to the diagram.

Line 376. Please use "dispensable" as opposed to "not indispensable".

Line 380. "bilateral cleavages decreased more than unilateral cleavages". Since the data appears to show that cleavage can occur at single sites and is thus uncoupled, it is not particularly interesting that bilateral cleavage decreases more than unilateral cleavage in presence of YY1. This is a straightforward consequence of independent inhibition of cleavage at either of the two TR5 elements. It is one thing to say that YY1 can increase TIR-TIR joining while suppressing transposition. But the

authors conclude that YY1 binding to TR5 is essential for TIR-dependent DNA recombination (Fig. 3) while at the same time showing that YY1 inhibits TIR cleavage (Fig. 5f-k). This is a fundamental contradiction that is not addressed by the authors and that undermines their conclusions.

Fig S7. Why does bbYY1 interact only with bbRAG2L in A,B,C but with bbRAG1L and bbRAG2L in F? The figure legend indicates a "surface cartoon structure" but the authors actually show two structures, one surface, one ribbon. What is the rationale?

The discussion is generally good but lines 442-453 and 492-502 are irrelevant for this work and the discussion would be much better focused without these sections.

Reviewer #3 (Invertebrate immune, VDJ, non-mouse/human immune) (Remarks to the Author):

This manuscript by Liu et al. is centered on studying the regulation of the ProtoRAG transposon in *Branchiostoma belcheri*. Using classical reporter assays the authors identify a transcriptional control element in the TIRs of the protoRAG transposon that corresponds to the consensus binding site for the transcription factor YY1. The authors then use a large array of in vitro and cell line based assays to show that YY1 can bind to these sequences, and by doing so YY1 appears to influence the efficiency and outcome of various steps of the transposition reaction. While the data looks largely solid, the manuscript suffers from being an overwhelming collection of data from countless experiments, and falls short of focusing on the key central findings and messages that the authors want to convey. In short, sometimes less is more. The main message of this manuscript is that YY1 seems to be a negative regulator of protoRAG transposition in the cell line based assays. What is really missing is some direct link to the biology of protoRAG transposition in *B. belcheri* (the only true in vivo system in which these events occur) – but may be this is not what the authors really intend to focus on (there seems to be trend towards wanting to stretch the data towards understanding RAG1/RAG2 biology in vertebrates). In summary, this manuscript provides potentially important insight into the regulation of protoRAG transposition. In its current state, however, this study is largely of interest for scientist in the field of mobile DNA elements, but not for the broad audience of Nature Communications. But rewriting and refocusing the data presented in this manuscript (and omitting some less important data) has the potential to reveal a clear message that could be a broad interest.

Major points

- 1) The authors study a single protoRAG element that was identified in the genome of *B. belcheri*. As this transposon is only weakly active (is there any quantification of what this weakly refers to) in vivo, it raises the question whether the element is fully functioning (and just suppressed by host factors) or already mutated to reduce its activity. Now here comes the idea of the authors into play that it may be transcriptional control elements that are important for the suppression of the activity. This could be used as a starting point to tell the story that the authors want to report here.
- 2) The reporter assays in Figure 1 are focused on identifying a region that positively controls transcription of the protoRAG transposon, but they end up identifying a factor that suppresses transcription. Surprisingly this conundrum is never discussed or explained. There might be even stronger repressors in the other parts of the transposon that are excluded from studies early on. Similarly, it is quite puzzling that no positive regulator of transcription was identified.
- 3) In the discussion the authors focus on trying to draw parallels between the regulation of protoRAG transcription and RAG1/RAG2 regulation – given the absence of TIR like elements in the vertebrate RAG1/RAG2 loci this discussion is not helpful at all and detracts from the main message – instead similarities to the transcriptional control of well studied transposons should be discussed.
- 4) There are a large number of experiments dedicated to study the formation of HDJs and TTJs but the relevance of these joints for protoRAG transposition are unclear and should be better explained. It seems that these joints are studied because their equivalents (CJs and SJs) are highly relevant in the context of V(D)J recombination, but we should not forget that protoRAG is (still) just a selfish

transposon. As abortive transposition events are detrimental to the survival of the transposon they should be avoided by the transposon. That would mean YY1 might actually be "beneficial" for the transposon and not a host factor that is important for suppression of activity. Again I would recommend to focus on the central question "how is transposition controlled" and not "how can we link this to V(D)J recombination".

5) In Figure 1, it is unclear why a construct encompassing the 5'TIR all the way to the TSS (-307/+413) was not tested. In addition, the rationale for inverting the 5'TIR in the luciferase reporter constructs in Supplementary Fig1B should be explained. If the authors want them to resemble the original transposon they should be in the correct (non-inverted) orientation.

6) The phylogenetic tree in Suppl. Figure 2A does not show whether proteins are well-conserved or not (in fact no phylogenetic tree would do that) – to show this one needs to show sequence alignments.

7) Line 189-191/Fig 2f. The authors claim in the text that the two REPO domains are important – I can only see one REPO domain, and deletion of that has only a very modest effect. Am I missing something here?

8) Lines 225-233: This is one of the few examples where a new reporter construct/assay is introduced and not clear rationale give as to why. The HDJs could have been cloned from the experiments done with construct pTIRG8 in Figure 3c (by sorting GFP+ cells, they could have skipped the necessity for nested PCR) – the authors should explain why they did not do that.

9) Throughout the manuscript the author go back and forth between using hsYY1 and bbYY1. If the focus is to understand protoRAG activity in *B. belcheri* the authors should focus on bbYY1. Given that a domain of hsYY1 is missing in bbYY1, it is unclear what we will learn by looking at hsYY1.

10) The experiments testing a putative interaction between bbYY1 and Ku70/80 are not very informative, as only hsKu70/80 were used and not the homologs from *B. belcheri*. But the notion that bbYY1 and hsYY1 may indirectly control DNA repair by regulation of the expression levels of DNA repair factors should be considered and discussed.

11) The "shorten" 5'TIREs need to be described in more detail and a description of their sequence should be one of the main Figures. As this alternative cleavage site is so prominent one wonders what the biological relevance of this site is. Was it part of an "ancestral" highly active protoRAG transposon or is it part of a "decaying" transposon in *B. belcheri*?

12) Lines 306-323: This section about the "cryptic" TIRs is confusing. Without a schematic drawing what the authors mean by "cryptic TIR" it is impossible for the reader to understand. If the authors are simply referring to 3'TIR with a TR5 mutation the term "cryptic" TIR is misleading and "mutated TIR" is more accurate.

13) In Figure 5, the authors describe a clear effect of YY1 on the cleavage phase of the transposition reaction. In light of this the previous section about the regulation of TIR-dependent transposition is really hard to interpret – all effects could be simply the result of less efficient cleavage. Hence I recommend to shorten that section.

14) It is highly recommended to ask someone with knowledge in the field who is fluent in English to help edit and correct the language of the manuscript. This will dramatically increase the readability. Below are only a few of the language glitches I caught.

Minor points:

1) Line 125: are the authors referring to SV40 and CMV promoters or enhancers?

2) Line 127-128: Figure 1 shows that the TSS is outside the 5'TIRs (drawing does not match text)

3) In Figure 1 and at numerous locations in the text it reads "PloyA" instead of PolyA

4) Line 172-174 / Supplementary Fig 2D/E: the statement that bbYY1 expression is high during the entire lifetime is not reflected in the Figures – muscle clearly shows a much lower expression level than all other adult tissues.

5) Suppl. Figure 2C: in the deltaNTD samples there are green "cells/nuclei" without a DAPI signal –

this should be explained.

6) Suppl. Figure 3D: I have no idea what the consensus motif on the right of the figure indicates

7) Lines 220-222: The statement that the TR5 and YY1 are essential is too strong, there clearly are events in the absence of these elements

8) Lines 252-255: The authors need to explain better where the CJs come from – I assume they are the result of the V(D)J recombination assay.

9) Line 298-300: there are tons of experiments in the literature that mouse RAG1/RAG2 recognize 23-RSS (it is their cognate target after all)

10) Line 413: is should read "charged" instead of "electric".

Point by point responses to the referees' comments

Reviewers' comments:

Reviewer #1 (VDJ, RAG transposition) (Remarks to the Author):

"I liked this paper the first time around - it should be published.

The two recent Nature papers cited by the authors indicate the importance of the topic and the relevance of this work from the Dreyfuss lab. Publish after the minor corrections I suggested last time."

Response #1. We highly appreciated your comments. It greatly encouraged us. Additional corrections have been made according to the second and the third reviewers' comments/suggestions. We hope that our revised manuscript will still be satisfied by you.

Reviewers' comments:

Reviewer #2 (AID, VDJ recombination) (Remarks to the Author):

"These authors previously identified and characterized a ProtoRAG transposon in lancelet that is thought to represent an evolutionary precursor of the RAG recombinase system used by the adaptive immune system in jawed vertebrates. The ProtoRAG transposon is composed of genes encoding RAG1-like and RAG2-like proteins flanked by TIR sequences related to RSSs. Here they show that the TIR sequences contain promoter elements and binding sites for lancelet YY1, which represses ProtoRAG transcription, stimulates TIR-dependent excision and recombination, but suppresses TIR-dependent transposition in favor of TIR-TIR joining. YY1 functions in part by direct binding to bbRAG2L. They contend that YY1 regulation of ProtoRAG

activity is beneficial for genome stability and was an early step in domestication of this transposon for use in the vertebrate immune system. This is, in principle, an intriguing and important contribution. However there are many problems with the presentation and some experimental shortcomings and inconsistencies that must be satisfactorily resolved before this manuscript should be given further consideration.”

Response #2. We highly appreciated your positive and constructive comments on our manuscript. We have improved our manuscript according to your comments. Followings are our point by point responses.

Reviewers' comments:

“The manuscript is inexpertly written and in sufficiently poor English that it is sometimes very difficult to imagine exactly what the experiment is and what the authors are trying to say. They need to pay much greater attention to the presentation and work closely with native English speaking editors to improve the manuscript.”

Response #3. Thank you for your comments. We have revised our manuscript carefully and made it reviewed by an English native speaker. We are sure that the revised manuscript has been greatly improved and it should be much clearer and easier for the readers to follow. Hope that the revised manuscript can meet your requirements.

Reviewers' comments:

The title is a bit misleading, as there is nothing in this manuscript about lymphoid restriction. I would suggest either “...evolutionary beginning of the vertebrate RAG system” or “...evolutionary beginning of domestication of the vertebrate RAG system”.

Response #4. Thank you for your constructive comments. We have revised the title as “Functional suppression of ancestral RAG transposon by YY1 revealed the initial domestication of the vertebrate RAG system”.

Reviewers' comments:

The discussion of RAG gene regulation in the 3rd paragraph of the Introduction (particularly 87-92) seems arbitrary and simplistic. With everything known about RAG gene regulation, why selectively call out NFATc1? What is the point?

Why selectively call out the role of distal elements 5' of RAG2? Why, for the sake of their argument, is it relevant that the elements are asymmetrically disposed? In fact, it is not clear from the writing what the authors mean by "asymmetrically cis genetic elements" if one did not know the literature already. Moreover, the obscure reference to region II will be understood by no one and will unnecessarily force readers to the literature.

Response #5: Thank you for pointing out these questions. Refer to your and other reviewers' comments, we have simplified this section to avoid the discussion about lymphoid restriction of RAG system. Please refer to paragraph 3 in introduction section (Page 5).

The figure legends, in general, are incorrectly written. They often state the result of the experiment, but do not always state what was actually done in the experiment. For example, Fig 1 b,d,e should mention that these are luciferase experiments. See also Fig. 2d,f. Correct as necessary throughout.

Fig 1b,d,e. No information is provided about replicates or statistical significance. Same is true for Fig S2D, S3D

Fig 1b. Luciferase orientation is depicted in Fig S1B but not here. Please be consistent.

Fig 1f. Please correct PolyA spelling.

Response #6. Thank you for your suggestions and corrections. We have revised all the figure legends in order to make it clear on the experimental process and avoid repeating the results. Moreover, all the corrections have been addressed according to your comments.

Line 128. Both TSSs are said to be located within the TIRs, but Fig. 1G shows this to be true only for the 3' TIR. Please clarify.

Line 141-144. The authors argue that the TIR sequences drive transcription of the RAG1L and

RAG2L. Although they have demonstrated promoter activity in luciferase assays, whether these promoters drive transcription of RAG1L and RAG2L is speculative. Can the authors demonstrate RAG1L and RAG2L transcripts initiating at these locations?

Response #7. Thank you for your correction. It is true that the TSS of bbRAG2L located within the 3'TIRs, while the TSS of bbRAG1L was out of the 5'TIRs. We have corrected the description about TSSs location in our revised manuscript. Please refer to Fig 1g.

As you said, it is important to demonstrate the rare TSSs of bbRAG1L and bbRAG2L transcripts. As we have mentioned in the *Cell* paper [1], the transcriptions of bbRAG1L and bbRAG2L are extremely low, thus it is hard for us to obtain their 5'UTR using 5'RACE technology from cDNA library of lancelet. Since the BAC73 clone contains the whole coding sequences of both bbRAG1L and bbRAG2L genes, as a replacement, we have tried to identify the complete transcripts of the *bbRAG1L* and *bbRAG2L* genes by transfecting BAC73 clone into 293T cells. However, after a great effort, we still failed to obtain the 5'UTR sequence of bbRAG1L and bbRAG2L by RACE. Thus, in order to identify the protential TSSs of *bbRAG1L* and *bbRAG2L* genes, we have to transfect the pGL-R1-1 or pGL-R2-1 constructs which contain the full length 5'TIR and 3'TIR into 293T cells for substitution, and then performed the RACE assay to determine the transcriptional start sites of the reporter gene. Although we could not directly identify the transcriptional start sites of both *bbRAG1L* and *bbRAG2L in vivo*, the replacement assays can also help us to identify the protential TSSs of *bbRAG1L* and *bbRAG2L* genes. We have stated clearly that how the TSSs of *bbRAG1L* and *bbRAG2L* were predicted in the revised manuscript. Hope that our explanation and revision can be accepted by you.

Fig 1h and all subsequent non-quantitative figures. The authors do not indicate whether the experiment was repeated with similar results.

Response #8. We appreciated your comments. Yes, all the non-quantitative figures were repeated by independent experiments at least twice and similar results have been obtained. We have made such statements in the relative figure legends.

Fig 2c. The fact that YY1 is inhibitory to transcription but mutation of the YY1 site causes reduced expression are, at face value, contradictory. The authors do not acknowledge this, leading to confusion. If the presumption is that YY1 competes with an unknown activator for this site, please state this hypothesis. Alternatively, YY1 could be an activator and the effect of YY1 knockdown could be an indirect effect.

Response #9. Thank you for your comments and nice suggestions. Yes, YY1 may cooperate with some co-repressors or activators to regulate the transcription of target genes as many studies have reported [2-5]. Here, the mechanism of how YY1 suppressed the transcription of *bbRAGL* is complicated and still required further systematic research, maybe by competing with an unknown activator for its binding site as you have speculated. We have acknowledged this in our revised manuscript. Please refer to Page 11, Line 200-203.

Fig 2d. If this figure involves transfection of 293TshYY1 then the figure legend should say so. The authors should show levels of human YY1 protein in transfectants compared to 293T wild-type and knockdown.

Response #10. Thank you for your suggestions. According to your suggestion, WB of human YY1 protein in transfectants compared to 293T wild-type and knockdown have been shown in Fig 2d (here indicated as Fig R1). The related statement was supplemented in the figure legend.

Fig R1. Luciferase reporter assays in 293T and 293TshYY1 cells indicated that both of bbYY1 and hsYY1 can suppress the transcriptional activity of *ProtoRAG*. Upper, the quantification of transcriptional activities of *pdTIR*. Lower, western blotting of YY1 transfectants.

Fig 2f. bbYY1 deletion mutants should be tested for binding to TIR DNA.

Response #11. We highly appreciated for this suggestion. We have tested the bindings of bbYY1 deletion mutants to TIR DNA by EMSA. The results were shown at the supplementary Fig.3E (here indicated as Fig R2) in our revised manuscript.

Fig R2. Representative EMSA gels showed that the ZNF domain is necessary for the binding of bbYY1 truncates to biotin labeled TIR probe. (A), Representative EMSA to show the binding of bbYY1 truncates to labeled oligomer 1. (B), SDS-PAGE of the GST tagged bbYY1 truncated proteins.

Fig 3b. I do not see where the authors describe precisely the mutations tested in the different constructs.

Response #12. Thank you for your constructive comment. We are sorry for the confusion. More descriptions of the mutated constructs have been added in Figure 3b and the related legends accordingly. Please refer to Page 12, Line 221-223 and Page 55,1097-1101.

Line 259. "reduce shortness of TIR dependent excision"? Please explain. Imprecision in coding joints appears to be attributed to altered specificity of cleavage but it remains possible that this represents an NHEJ defect. This should be acknowledged.

Response #13. Thank you for your question and constructive comment. As you have mentioned, the imprecision in HDJs may be attributed to altered specificity of cleavage. It may also be due to the NHEJ defect. We have acknowledged this in our revised manuscript. Please refer to Page 14-15, Line 279-287.

Fig 4a, legend to Fig 4, "shorten" is inappropriate usage. Heading Fig 4b by "rTaq Mix: 35 cycles" and Fig 4c by "KOD plus neo: 45 cycles" is confusing because it is not mentioned in the figure legend. Are the names of the polymerases rTaq mix and KOD plus neo? The Methods indicate the former at 30 cycles but the figure says 35. Please explain. Use the figure legend to explain the experiment. If 4b and 4c are essentially repeats of the same experiment, the recovery of short TIRs is not reproducible. Has this experiment been repeated sufficiently to know if the results are consistent and that the quantitative differences are statistically significant? Is there a control PCR to normalize 293T to 293TshYY1? Conclusions about the influence of YY1 on cleavage need strengthening.

Response #14. We highly appreciated for pointing out these problems in detail. The inappropriate usage of words in the figure and figure legend have been corrected. The figure legend and method section have also been revised correctly according to your suggestions.

This LM-PCR experiment has at least been repeated for three times. A control PCR of the expression of *Kanamycin* resistance gene on *pTIR104* was performed to normalize 293T to 293T^{shYY1} in these LM-PCR assays. Following is one of our repeated results (provided as Fig 4 in our revised manuscript, here indicated as Fig R3).

Fig R3. (A), Illustration of the LM-PCR method for detecting cleavage sites of TIR substrate after bbRAG1L/2L mediated recombination. As illustrated, if the cleavages happened at the borders of both 5' and 3'TIR, a right TIREs would be amplified by LM-PCR using specific primer pairs. In contrast, if the cleavage happened between the 5' and 3'TIR, short TIREs would be amplified by LM-PCR. These TIREs products were finally sequenced to determine the cleavage sites. P5, P6 and P7 indicate specific PCR primers.

(B) and (C), Representative LM-PCR gels showed less and short TIREs occurred after TIR-dependent recombination in 293T^{shYY1} cells. sTIRE is referred to short TIR ends. The expression of normalized control gene (*Kanamycin* resistance gene, indicated as *Kan*) on *pTIR104* was also shown. LM-PCRs were performed using rTaq DNA polymerase with 30 cycles or the KOD plus neo DNA polymerase with 45 cycles, respectively. The representative LM-PCR gels represents three independent replications.

Fig S4GLines 291-3. By "aimed DNA recognizing" do you mean "target DNA recognition"? Since you have already concluded that YY1 promotes more abundant and more precise recombination and cleavage, what is the rationale for examining whether "YY1 deficiency

would shift recognition sites of bbRAGL to TIRs". This does not seem to be sensible.

Response #15. Thank you for your comments and helpful suggestions. We have corrected the inaccurate description accordingly. As we have explained above, YY1 promotes more precise recombination, probably by helping the specific targeting of bbRAGL to TIRs. As we known, the cleavage should happen after DNA recognition, nicking and breaking. Thus, it is important to reveal "whether YY1 would influence the recognition of bbRAGL to TIRs". As Supplementary Figure 6D, E in revised manuscript showed, bbYY1 indeed benefits to the recognition of bbRAGL to TIRs, suggesting that YY1 can promote precise recombination by helping precise targeting. Hence, we thought that such test is meaningful. In order to avoid misleading, we have revised the word "shift" as "affect" in the revised manuscript.

Lines 298-301. Everything about this sentence is incorrect: "However, unlike mouse RAG1/RAG2 complex, bbRAG1L/2L could recognize 23-RSS" – it should be: "like mouse RAG1/RAG2". and then "indicating that the binding of bbRAG1L/2L complex to 23-RSS DNA sequence is not species specific" – but this makes no sense since there is only one 23-RSS tested (mouse). There is no 23-RSS in lancelet. Please correct the first part of the sentence and there is no need to say anything further.

Response #16. Thanks for the correction and suggestion. This sentence has been revised accordingly as "The bbRAG1L/2L complex could also recognize 23-RSS, indicating that bbRAG1L/2L may bind to an unspecific DNA *in vivo*, which should benefit transposition targeting." in our revised manuscript. Please see page 16, Line 321-323.

Figs S5G,H are uninterpretable. The text (304-305) describes a result assessing binding of bbRAG1L/2L to wild-type TIRs in presence of YY1. But the figure panels compare wild-type to mutant TIRs and there is no indication as to whether YY1 is transfected in all conditions or only in some. And the figure legend describes a different result that is never mentioned in the text, regarding a comparison of wild-type to mutant. Unfortunately the figure legend does not provide basic information on the nature of the experimental manipulation. Please

synchronize the text, the figure and the figure legend so this becomes coherent. I can draw no conclusions.

Response #17. Thank you for your corrections. We are sorry for the confusions caused by the misleading expression in the main text. The misleading sentence have been rephrased as "The results showed that more bbRAG1L/2L and bbYY1 were pulled down by WT TIRs but not the TR5 mutated TIR (Bio-5' TIR-Mu) (Supplementary Fig. 6D, E), indicating that bbYY1 may help bbRAGL to recognize TIR specifically." Besides, clearer figure legend and corresponded text have also been well synchronized according to your suggestions. Please refer to page 16, Line 326-328.

Lines 318-321. Why is the ratio of GFP+ cells increased by YY1 if TR5 is mutated? In addition, it is stated that "the ratio of mCherry+ cells increased more than that of GFP+ cells when bbYY1 or hsYY1 was overexpressed" even though this does not appear to be true for the latter. Finally, Fig 4e, although performed three times, does not provide any statistical assessment of significant differences.

Response #18. Thank you for your question. The figure you referred have been re-drawn and statistical assessment of significant differences were provided in our revised manuscript. Please see the Fig 4f in revised manuscript (here indicated as Fig R4). It is true that the ratio of GFP+ cells increased by YY1 *in vivo*. Besides, the inaccuracy expression that you have pointed out has been corrected as "Meanwhile, both ratios of GFP and mCherry-positive cells were increased when bbYY1 or hsYY1 was overexpressed in 293T^{shYY1} cells. Moreover, the ratio of mCherry-positive cells increased to a greater extent than that of GFP-positive cells when bbYY1 was over-expressed (Fig. 4f and Supplementary Fig. 6F)" in the revised manuscript.

We tried to explain why the GFP positive cells increased as follows:

First, DNA pulldown evidences have indicated that bbYY1 benefits the recognition of bbRAGL to intact TIR. The ratio of GFP+ cells increased by YY1 may be an indirect effect of that more bbRAGL were recruited to TIR when YY1 was overexpressed *in vivo*.

Second, since YY1 could help DNA repairing [6-8], another possibility is that YY1 may prompt DNA repairing after the cleavage step, which may also lead to higher ratios of GFP+ cells when YY1 was overexpressed *in vivo*.

Fig R4. FACS assay indicated that YY1 contributes to the recognition of right TIR *in vivo*. (A) Schematic diagram illustration of different TIR recognitions mediated by bbRAG1L/2L *in vivo*. The recognition and cleavage happened between sites 1 and 2, or between sites 1 and 3, would lead to GFP or mCherry expression respectively. (B) Percentage of GFP and mCherry positive cells after the bbRAG1L/2L mediated recombination in 293T and 293T^{shYY1} cells. Values stand for mean \pm SD, with n = 3. A two-tailed, unpaired Student's *t*-test was used for comparisons. *P < 0.05; **P < 0.01; ***P < 0.001; ****P < 0.0001. ns, no significant.

Fig 5d. Please add CptG and TIRG10 to the diagram.

Line 376. Please use "dispensable" as opposed to "not indispensable".

Response #19. Thank you for your nice suggestions. We have added necessary description of CptG and TIR10 in the figure and revised the text accordingly. The inaccurate word has also been revised.

Line 380. "bilateral cleavages decreased more than unilateral cleavages". Since the data appears to show that cleavage can occur at single sites and is thus uncoupled, it is not

particularly interesting that bilateral cleavage decreases more than unilateral cleavage in presence of YY1. This is a straightforward consequence of independent inhibition of cleavage at either of the two TR5 elements.

Response #20. Thank you for your comments. We have corrected this description in our revised manuscript according to your comment. Please see page 20, Line 402-405.

It is one thing to say that YY1 can increase TIR-TIR joining while suppressing transposition. But the authors conclude that YY1 binding to TR5 is essential for TIR-dependent DNA recombination (Fig. 3) while at the same time showing that YY1 inhibits TIR cleavage (Fig. 5f-k). This is a fundamental contradiction that is not addressed by the authors and that undermines their conclusions.

Response #21. Thank you for your comments. The observation that YY1 can increase TIR-TIR joining (TTJ) while suppress transposition, we think, is not opposite. Although DNA recombination and transposition were both depended on the same cleavage step, they were two different events after TIR DNA cleavage. Be similar to SJs formation in V(D)J recombination, efficient TTJ would benefit to the TIR-TIR rejoining to avoid the explosion of its attractive terminals, resulting in the reduction of TIR-TIR mediated transposition. Thus, it is understandable that YY1 increased TIR-TIR joining while suppressed transposition *in vivo*. Moreover, the finding that YY1 benefits to the TTJ rejoining is in line with that it benefits the HDJ recombination after TIR cleavage. The promotion of both TTJ and HDJ formation indicated the roles of bbYY1 in DNA repair progress.

Fig S7. Why does bbYY1 interact only with bbRAG2L in A,B,C but with bbRAG1L and bbRAG2L in F? The figure legend indicates a "surface cartoon structure" but the authors actually show two structures, one surface, one ribbon. What is the rationale?

Response #22. Thank you for your questions. Our previous study has shown that bbRAG1L interacts with bbRAG2L for full function [1]. We also found that bbRAG2L is more stable when

it is co-expressed with bbRAG1L, which benefits the detection between bbYY1 and bbRAGL complex. To obtain a direct evidence, an additional GST pulldown assay to detect the direct interaction between cZNF mutants and single bbRAG2L was provided as Supplementary Fig. 8E in our revised manuscript (here indicated as Fig R5). For the description of "surface cartoon structure", we have corrected this inaccuracy expression as "the surface and the cartoon structure" in our revised manuscript.

Fig R5. GST-pulldown assay showed that the mutation of negative charged residues on coils (between α -helices and β -sheets) of core zinc finger (cZNF) destroyed the interaction between cZNF and bbRAG2L.

The discussion is generally good but lines 442-453 and 492-502 are irrelevant for this work and it the discussion would be much better focused without these sections.

Response #23. We highly appreciated your helpful suggestions. We have revised the discuss section accordingly.

Reviewer #3 (Invertebrate immune, VDJ, non-mouse/human immune) (Remarks to the Author):

This manuscript by Liu et al. is centered on studying the regulation of the ProtoRAG transposon in *Branchiostoma belcheri*. Using classical reporter assays the authors identify a transcriptional control element in the TIRs of the protoRAG transposon that corresponds to the consensus binding site for the transcription factor YY1. The authors then use a large array of in vitro and cell line based assays to show that YY1 can bind to these sequences, and by doing so YY1 appears to influence the efficiency and outcome of various steps of the transposition reaction. While the data looks largely solid, the manuscript suffers from being an overwhelming collection of data from countless experiments, and falls short of focusing on the key central findings and messages that the authors want to convey. In short, sometimes less is more. The main message of this manuscript is that YY1 seems to be a negative regulator of protoRAG transposition in the cell line based assays. What is really missing is some direct link to the biology of protoRAG transposition in *B. belcheri* (the only true in vivo system in which these events occur) – but may be this is not what the authors really intend to focus on (there seems to be trend towards wanting to stretch the data towards understanding RAG1/RAG2 biology in vertebrates). In summary, this manuscript provides potentially important insight into the regulation of protoRAG transposition. In its current state, however, this study is largely of interest for scientist in the field of mobile DNA elements, but not for the broad audience of Nature Communications. But rewriting and refocusing the data presented in this manuscript (and omitting some less important data) has the potential to reveal a clear message that could be a broad interest.

Response #24. Thank you for your summarization and constructive suggestions. We have revised the manuscript according to your suggestions point by point in order to make it more logical and clearer for readers to follow. We also have made it read through by an English native speaker. We are sure that our revised manuscript has been greatly improved and hope that all the efforts of revisions can be accepted by you.

Major points

1) The authors study a single protoRAG element that was identified in the genome of *B. belcheri*. As this transposon is only weakly active (is there any quantification of what this weakly refers to) in vivo, it raises the question whether the element is fully functioning (and just suppressed by host factors) or already mutated to reduce its activity. Now here comes the idea of the authors into play that it may be transcriptional control elements that are important for the suppression of the activity. This could be used as a starting point to tell the story that the authors want to report here.

Response #25. We highly appreciate your constructive suggestions. We have reorganized our starting point of the story in the revised manuscript according to your suggestion.

2) The reporter assays in Figure 1 are focused on identifying a region that positively controls transcription of the protoRAG transposon, but they end up identifying a factor that suppresses transcription. Surprisingly this conundrum is never discussed or explained. There might be even stronger repressors in the other parts of the transposon that are excluded from studies early on. Similarly, it is quite puzzling that no positive regulator of transcription was identified.

Response #26. Thank you for your comments. There are two aims to start this study, one is to identify something that may lead to the weak expression of bbRAGL complex, and the other is to reveal how the active *ProtoRAG* transposon was strictly controlled by the host to maintain the genome stability of the host. Guided by the first question, we have to identify the region (*cis*-element) that controls the transcription of *ProtoRAG* and finally found the core *cis*-elements that control the transcription within or nearby the TIR sequences by a series of reporter assays. Then, the *trans*-factor YY1 was identified to bind to the core *cis*-elements to suppress the transcription of *bbRAG1L* and *bbRAG2L*. Thus, it is reasonable and logical. Besides YY1, some other factors, such as Arnt, FOXI, E2F, ZBTB33 *et al* were also predicted to bind the core *cis*-elements, which were referred in the revised manuscript. Some of them may positively regulate the transcription of bbRAGL complex, but it is not the topic of this

manuscript.

3) In the discussion the authors focus on trying to draw parallels between the regulation of protoRAG transcription and RAG1/RAG2 regulation – given the absence of TIR like elements in the vertebrate RAG1/RAG2 loci this discussion is not helpful at all and detracts from the main message – instead similarities to the transcriptional control of well studied transposons should be discussed.

Response #27. Thank you for your constructive suggestion. According to your suggestion, we have reduced the section of discussion on parallels of transcriptional regulation between *ProtoRAG* and mouse RAG1/RAG2. We have also added more discussion on the similarities of transcriptional control between well-studied transposons and *ProtoRAG* in our revised manuscript. Please see page 23-24, Line 470-476, Line 491-499.

4) There are a large number of experiments dedicated to study the formation of HDJs and TTJs but the relevance of these joints for protoRAG transposition are unclear and should be better explained. It seems that these joints are studied because their equivalents (CJs and SJs) are highly relevant in the context of V(D)J recombination, but we should not forget that protoRAG is (still) just a selfish transposon. As abortive transposition events are detrimental to the survival of the transposon they should be avoided by the transposon. That would mean YY1 might actually be “beneficial” for the transposon and not a host factor that is important for suppression of activity. Again I would recommend to focus on the central question “how is transposition controlled” and not “how can we link this to V(D)J recombination”.

Response #28. Thank you for your constructive comments. In the present study, YY1 benefits to precise HDJ formation and precise cleavage, helping to maintain the genome stability of host. However, it can also suppress the transposition of *ProtoRAG* by improving the TIR-TIR rejoining after transposon excision. Thus, bbYY1 may be a double-edged sword that is beneficial for the transposon but also a host factor that is important for suppression of transposon activity. In addition, as you suggested, we have focused more on the regulation

of *ProtoRAG* itself to avoid misleading.

5) In Figure 1, it is unclear why a construct encompassing the 5'TIR all the way to the TSS (-307/+413) was not tested. In addition, the rationale for inverting the 5'TIR in the luciferase reporter constructs in Supplementary Fig1B should be explained. If the authors want them to resemble the original transposon they should be in the correct (non-inverted) orientation.

Response #29. Thank you for pointing out these problems. Why we previously inverted the 5'TIR in luciferase reporter constructs is due to the consideration that the activity of the inverted orientation of TIRs was higher than that of the non-inverted ones, which would benefit to the identification of the core *cis*-elements. Now, data of construct containing (-307/+413) of 5'TIR (provided as Fig 1b in the revised manuscript, here indicated as Fig R6_A) and new data of non-inverted TIRs (provided as Supplementary Fig. 1B in the revised manuscript, here indicated as Fig R6_B) were both provided in the revised manuscript accordingly. Hope that these revisions can meet your requirements.

Fig R6_A. Luciferase reporter experiments showed that 5' TIR-FL of *ProtoRAG* contain *cis*-acting elements that may drive the transcription of *bbRAG1L*.

Fig R6_B. Luciferase reporter experiments showed that the -307 to -238 bp upstream of *bbRAG1L* TSS may be regarded as the core elements for the transcription of *bbRAG1L*.

6) The phylogenetic tree in Suppl. Figure 2A does not show whether proteins are well-conserved or not (in fact no phylogenetic tree would do that) – to show this one needs to show sequence alignments.

Response #30. Thank you for your comments. Sequence alignments of YY1 homologues of representative species have been added in our revised manuscript accordingly. Please refer to the supplementary Figure 2. (Here, indicated as Fig R7).

Fig R7. Alignment of YY1-like proteins from representative species showed certain conservative domain on bbYY1. Dashed box indicated domains that exist in human but lack in lancelet. Conserved sequences were red colored. The accession numbers for YY1-like proteins include

NP_003394.1 (human), NP_033563.2 (mouse), XP_028916341.1 (platypus), XP_027313566.1 (duck), NP_997782.1 (zebrafish), NP_001087404.1 (frog), XP_007886520.1 (shark), XP_002610706.1 (*Branchiostoma floridae*), MF966513-MF966514 (*Branchiostoma belcheri*).

7) Line 189-191/Fig 2f. The authors claim in the text that the two REPO domains are important – I can only see one REPO domain, and deletion of that has only a very modest effect. Am I missing something here?

Response #31. Thank you for your questions. We are sorry for the inaccurate and misleading expression. Now we have corrected this sentence as “The results showed that the conserved REPO and ZNF domains on bbYY1 are important for its transcriptional inhibition of reporter genes (Fig. 2g).” in our revised manuscript. Please see page 11, Line 206-209.

8) Lines 225-233: This is one of the few examples where a new reporter construct/assay is introduced and not clear rationale give as to why. The HDJs could have been cloned from the experiments done with construct pTIRG8 in Figure 3c (by sorting GFP+ cells, they could have skipped the necessity for nested PCR) – the authors should explain why they did not do that.

Response #32. Thank you for your comments. We are sorry for the confusion. A clear illustration of different substrates was provided in the revised manuscript (Figure 1f). In brief, the *pdTIR* contains a pair of full length TIRs. However, the *pTIRG8* and *pTIR104* have a pair of mini-core TIRs that were named as minimal TIRs and found to mediate efficient recombination in our previous article [1]. For better distinguish of TIRs on different constructs, the TIRs on *pdTIR* and *pTIR104* were signaled as TIR-FL and TIR respectively. Reasons of using the *pdTIR* construct is that the recombination using TIR-FL is more efficient than minimal TIRs. Thus, it would be more sensitive for our PCR amplification. We also wanted to explain why two iterations of PCR assays were performed. As the supplementary figure 5C in our revised manuscript illustrated, the first and the second time of HDJ PCR products were different.

9) Throughout the manuscript the author go back and forth between using hsYY1 and bbYY1. If the focus is to understand protoRAG activity in *B. belcheri* the authors should focus of bbYY1. Given that a domain of hsYY1 is missing in bbYY1, it is unclear what we will learn by looking of hsYY1.

Response #33. Thank you for your comments. Considering that most of the assays were performed in 293T cells, the comparisons between bbYY1 and hsYY1 were retained in the revised manuscript in order to make the manuscript more logical. Since the DNA binding domain is highly conserved between hsYY1 and bbYY1, parallel comparison of bbYY1 and hsYY1 would make it easier to understand why both bbYY1 and hsYY1 can bind to TIRs of *ProtoRAG*. However, as you have mentioned, there are some differences between bbYY1 and hsYY1, such as the lack of transcriptional activation domain in bbYY1. This domain difference is in consist with the result that bbYY1 inhibits but not induces the transcription of *ProtoRAG*. According to your comments, more clear description of the similarity and difference between bbYY1 and hsYY1 were added in our revised manuscript to make it easy to follow.

10) The experiments testing a putative interaction between bbYY1 and Ku70/80 are not very informative, as only hsKu70/80 were used and not the homologs from *B. belcheri*. But the notion that bbYY1 and hsYY1 may be indirectly control DNA repair by regulation the expression levels of DNA repair factors should be considered and discussed.

Response #34. Thank you for your comments. Since how YY1 involved in the DNA repair pathway is much complicated and required further analyzed, the result of putative interaction between bbYY1 and Ku70/80 were omitted in our revised manuscript. More discussion about YY1 involved in DNA repair has been added in our revised manuscript. Please refer to page 14, Line 279-287.

11) The "shorten" 5'TIREs need to be described in more detail and a description of their sequence should be one of the main Figures. As this alternative cleavage site is so prominent

one wonders what the biological relevance of this site is. Was it part of an “ancestral” highly active protoRAG transposon or is it part of a “decaying” transposon in *B. belcheri*?

Response #35. Thanks for the constructive suggestions. We have added more details of the “short” 5'TIREs accordingly in our revised manuscript. In addition, a described figure of the “short” 5'TIREs sequences has been added in Figure 4d of the revised manuscript. As Figure 4b and c showed, the alternative cleavage site is not a defined site, but it in fact represents many alternative cleavage sites, which exhibited as smear bands after gel separation (you could see the above mentioned Fig R3_B and C in this document). Imprecise cleavages happened at these alternative sites might lead to the loss of the intact transposon gene structure, and the formation of a “decaying” transposon in *B. belcheri*. However, we inferred that this situation perhaps rarely happened because of the high abundance of bbYY1 in lancelet.

12) Lines 306-323: This section about the “cryptic” TIRs confusing. Without a schematic drawing what the authors mean by “cryptic TIR” it is impossible for the reader to understand. If the author are simply referring to 3'TIR with a TR5 mutation the term “cryptic” TIR is misleading and “mutated TIR” is more accurate.

Response #36. Thank you for your comments. We are sorry for the confusing of cryptic TIRs. The “cryptic” was a word that firstly used in the published article by Lewis SM et al [9]. It means some DNA sequences which were resemble to authentic joining signals, but sometimes rearranged by mistakes. Since the *pSel G-mCh* and *pCptG* contain both of the WT 5'TIR and the TR5 mutated 3'TIR, we referred this abnormal TIR pair as cryptic TIRs previously. Here in the revised manuscript, we have referred them as mutated TIR according to your suggestion.

13) In Figure 5, the authors describe a clear effect of YY1 on the cleavage phase of the transposition reaction. In light on this the previous section about the regulation of TIR-dependent transposition is really hard to interpret – all effects could be simply the result

of less efficient cleavage. Hence I recommend to shorten that section.

Response #37. Thank you for your nice comments. We agreed with you that it is hard to interpret the function of YY1 on the regulation of TIR-dependent transposition. As per your suggestion, we have simplified this section just mentioning that bbYY1 may inhibit the transposition possibly by regulating the cleavage step due to a series of *in vitro* cleavage assays. YY1 may also help the DNA repair process to benefit the TTJ rejoining, leading to the suppression of *ProtoRAG* transposition.

14) It is highly recommended to ask someone with knowledge in the field who is fluent in English to help edit and correct the language of the manuscript. This will dramatically increase the readability. Below are only a few of the languages glitches I caught.

Response #38. Thank you very much for your kind comments. We have revised our manuscript and made it reviewed by an English native speaker. We are sure that the revised manuscript has been greatly improved and would be more clear and easier to follow. We hope that the revised manuscript can meet your requirements.

Minor points:

1) Line 125: are the authors referring to SV40 and CMV promoters or enhancers?

Response #39. Thank you for your question. Here we are referring to SV40 and CMV as promoters. The related corrections have been made in the revised manuscript accordingly.

2) Line127-128: Figure 1 shows that the TSS is outside the 5'TIRs (drawing does not match text)

Response #40. Thank you for pointing out this inaccurate description. As you have pointed out, one TSS located within the 3'TIR, and the other located nearby the 5'TIR. We have corrected the description about TSSs location in our revised manuscript. Please see page 8,

Line 129-135, and Figure 1g.

3) In Figure 1 and at numerous location in the text it reads "PloyA" instead of PolyA

Response #41. Thank you for pointing out this spelling error. All the spelling errors have been fixed in the revised manuscript accordingly.

4) Line 172-174 / Supplementary Fig 2D/E: the statement that *bbYY1* expression is high during the entire lifetime is not reflected in the Figures – muscle clear shows a much lower expression level than all other adult tissues.

Response #42. Thank you for your question. We are sorry for the misleading of the previous supplementary Fig 2D (This figure has been corrected and now provided as Supplementary Fig 3C in the revised manuscript) intended to show the relative expression of *bbYY1* normalized to *bbGAPDH*. As it shows, the fold value of *bbYY1* expression normalized to that of *bbGAPDH* is about 0.1 in muscle, and the expression of *bbYY1* in other tissues is comparative and highly similar to *bbGAPDH* which is a housekeeping gene, leading to the statement that "*bbYY1* expression is abundant during the entire lifetime". Moreover, due to our observations, many of the genes in lancelet have relative low expression in muscle without exception of *bbYY1*.

5) Suppl. Figure 2C: in the deltaNTD samples there are green "cells/nuclei" without a DAPI signal – this should be explained.

Response #43. Thank you for your comment. A new figure with clear DAPI staining has been added in the revised Supplementary Figure 3A, (Here indicated as Fig R8).

Fig R8. Laser confocal images of bbYY1 truncates suggested a nucleus location signal on the zinc finger domain of bbYY1.

6) Suppl. Figure 3D: I have no idea what the consensus motif on the right of the figure indicates

Response #44. Thank you for your question. We have added necessary description of the consensus motif in the Figure legends of Supplementary Figure 4D in our revised manuscript. The bits/height of each capital letter represents conservation of nucleotide on YY1 binding motif. Mutations of these conservative nucleotides would greatly decrease ratios of GFP-positive cells, suggesting a relationship between the conserved nucleotide on YY1 binding motif and the efficiency of bbRAGL mediated DNA recombination.

7) Lines 220-222: The statement that the TR5 and YY1 are essential is too strong, there clearly are events in the absence of these elements

Response #45. Thank you for your comments. We have revised the sentence as "Collectively, these data suggested that both the 9-bp TR5 element on *ProtoRAG* and *trans*-factor bbYY1 (or hsYY1) are important elements for TIR-dependent recombination mediated by bbRAG1L/2L.". Please see page 13, Line 242-244.

8) Lines 252-255: The authors need to explain better where the CJs come from – I assume they are the result of the V(D)J recombination assay.

Response #46. Thank you for your constructive suggestion. Yes, the CJs came from a V(D)J recombination assay. Since CJs formation was resemble to that of HDJs, we made an analytical analogy between HDJs and CJs. We have added additional information of CJ formation in the revised manuscript. Please see page 14, Line 271-275.

9) Line 298-300: there are tons of experiments in the literature that mouse RAG1/RAG2 recognize 23-RSS (it is their cognate target after all)

Response #47. Thank you for pointing out the incorrect description. We have corrected it as "The bbRAG1L/2L complex could also recognize 23-RSS (Supplementary Fig. 6C), indicating that bbRAG1L/2L may bind to an unspecific DNA *in vivo*, which should benefit transposition targeting." in our revised manuscript. Please see page 16, Line 321-323.

10) Line 413: is should read "charged" instead of "electric".

Response #48. Thank you for your nice suggestion. We have replaced the word accordingly in our revised manuscript.

Reference

1. Huang, S., et al., Discovery of an active RAG transposon illuminates the origins of V(D)J recombination. *Cell*, 2016. **166**(1): p. 102-14.
2. Zhang, Q., et al., The oncogenic role of Yin Yang 1. *Crit Rev Oncog*, 2011. **16**(3-4): p. 163-97.
3. Shi, Y., J.S. Lee, and K.M. Galvin, Everything you have ever wanted to know about Yin Yang 1. *Biochim Biophys Acta*, 1997. **1332**(2): p. F49-66.
4. Shi, Y., et al., Transcriptional repression by YY1, a human GLI-Kruppel-related protein, and relief of repression by adenovirus E1A protein. *Cell*, 1991. **67**(2): p. 377-88.
5. Seto, E., Y. Shi, and T. Shenk, YY1 is an initiator sequence-binding protein that directs and activates transcription in vitro. *Nature*, 1991. **354**(6350): p. 241-5.
6. Wu, S., et al., Loss of YY1 impacts the heterochromatic state and meiotic double-strand breaks during mouse spermatogenesis. *Mol Cell Biol*, 2009. **29**(23): p. 6245-56.
7. Wu, S., et al., A YY1-INO80 complex regulates genomic stability through homologous recombination-based repair. *Nat Struct Mol Biol*, 2007. **14**(12): p. 1165-72.
8. Oei, S.L. and Y. Shi, Transcription factor Yin Yang 1 stimulates poly(ADP-ribosyl)ation and DNA repair. *Biochem Biophys Res Commun*, 2001. **284**(2): p. 450-4.
9. Lewis, S.M., et al., Cryptic signals and the fidelity of V(D)J joining. *Mol Cell Biol*, 1997. **17**(6): p. 3125-36.

REVIEWER COMMENTS

Reviewer #2 (Remarks to the Author):

Although still a challenge to get through, this version of the manuscript is improved substantially over the original.

I do still have one important unanswered question:

The authors have misunderstood or overlooked one of my comments in their response #21. My main point was that the authors show and conclude that YY1 is indispensable for TIR-dependent DNA recombination (see section beginning on line 214 and Fig 3 c,d). They also show that there are fewer TIR ends with reduced YY1 in vivo (see section beginning on line 290 and Fig 4 a,b,c), which is consistent with the effect on recombination. But they show that YY1 inhibits cleavage of TIR substrates in vitro (fewer ends with more YY1; see section beginning on line 384 and Fig 5 f,g). And they conclude that one mechanism by which YY1 reduces transposition is by inhibiting cleavage (see line 418). But how can all of this be true? Why are TIR ends increased by YY1 in vivo and reduced by YY1 in vitro. And how can reduced cleavage by YY1 be used as an explanation for reduced transposition when it would also lead to a reduction in recombination? The authors show YY1 to be indispensable for recombination. In my mind, these contradictions undermine the conclusions of the paper. They need to be resolved.

Reviewer #3 (Remarks to the Author):

While this manuscript has somewhat improved, my central concern remains: it is overly long and instead of telling one consistent and clear story it tries to overwhelm both the reader and the reviewers with too much data (8 supplemental figures) – but as one could see from the detailed comments of reviewer #1 on the previous round, there were tons of little issues with almost everyone of them. And within all these experimental findings there is unfortunately still very little support for their main conclusion that “... BbYY1 contributes to host domestication of the protoRAG transposon ...”. There is no evidence anywhere that either the TIRs or BbYY1 evolved in a certain way to allow for these observed variety of “regulatory modes” (only TIRs of one ProtoRAG fragment are tested, and seemingly HsYY1 works just as well as BbYY1) – it may as well be that both, the ProtoRAG transposon and BbYY1, have co-existed in their current form over more 100 millions of years. In the last round I suggested to focus on the “regulation” as a theme, but the authors still tried to push too much in the direction of “domestication”.

To illustrate, I pick three examples where even the revised version falls short reaching the bar:

A) My previous concern 9 and the author’s response #33.

This manuscript is focused on understanding the biology of the ProtoRAG transposon in *B. belcheri*. So the relevant YY1 molecule to use for the studies is BbYY1. As no *B. belcheri* cell line exists, human 293T cells are used instead and the HsYY1 knock-down cell line reconstituted with BbYY1 is the appropriate testbed. Why is it important to use BbYY1? Because it differs in the central feature from HsYY1: it lacks the classical activation domain. So the rationale behind reporting lots of experiments with HsYY1 while an appropriate cell model with BbYY1 was available is unclear to me. Omitting some of this superfluous data would make the story much easier to understand.

B) My previous concerns 4 and the author’s response #28.

The rationale behind studying the HDJ formation (and even more so why this is done right after studying a role of BbYY1 in TIR-dependent DNA recombination) remains completely unclear – would one not first want to study transposition (as this is a transposon after all) when trying to understand transposon biology? The importance of HDJ in both *B. belcheri* biology and ProtoRAG transposon activity is unclear, the only point where it makes sense to study it is when trying to reveal similarities

and differences to vertebrate RAGs.

C) My previous concern 2 and the author's response #26.

Now that the authors state they also found other TF-binding sites (new here compared to the previous version). When trying to understand the biology of a transposon would one not start with the big effects (the 5'TIR boosts transcription 120-fold over background), yet here this study is centered on the two-fold decrease to what would be 60-fold (Figure 1) in transcriptional stimulation. One must simply wonder why the other factors were ignored (despite no-perfect matches of their binding sites) and whether those may have even bigger effects of ProtoRAG transposon behavior at all levels ranging from TIR recognition to transposition. I could not comment on this in the previous version as I did not know these facts. Now it makes me wonder whether these other factors are overall more important.

Point-by-point responses to the referees' comments

Reviewer #2 (Remarks to the Author):

Although still a challenge to get through, this version of the manuscript is improved substantially over the original.

I do still have one important unanswered question:

The authors have misunderstood or overlooked one of my comments in their response #21. My main point was that the authors show and conclude that YY1 is indispensable for TIR-dependent DNA recombination (see section beginning on line 214 and Fig 3 c,d). They also show that there are fewer TIR ends with reduced YY1 in vivo (see section beginning on line 290 and Fig 4 a,b,c), which is consistent with the effect on recombination. But they show that YY1 inhibits cleavage of TIR substrates in vitro (fewer ends with more YY1; see section beginning on line 384 and Fig 5 f,g). And they conclude that one mechanism by which YY1 reduces transposition is by inhibiting cleavage (see line 418). But how can all of this be true? Why are TIR ends increased by YY1 in vivo and reduced by YY1 in vitro. And how can reduced cleavage by YY1 be used as an explanation for reduced transposition when it would also lead to a reduction in recombination? The authors show YY1 to be indispensable for recombination. In my mind, these contradictions undermine the conclusions of the paper. They need to be resolved.

Response to Reviewer #2

Thank you very much for pointing out this question. We apologize for the misunderstanding of your comments during the first round of revision. As you have pointed out, the *in vivo* results indicated fewer TIR cleavages when hsYY1 was silenced, while *in vitro* cleavage assays showed contrasting results. We thought the appearance of such contradictory results might be due to the conditional difference between *in vivo* and *in vitro* experimental systems. A similar situation can also be found in previous research. For example, our recent study published in *National Science Review* showed that the TR5 element is significantly important for TIR cleavage in 293T cells¹. However, another *Nature* paper co-authored by David G. Schatz' and our group has found that the TR5 element is insufficient for cleavage of TIRs by the bbRAG1L/2L complex *in vitro*².

To avoid misleading the readers, we decided to remove the *in vitro* cleavage results from this revised manuscript due to the following consideration. First, experiments performed in 293T and 293T^{shYY1} cells provided consistent results, such as a reduced recombination efficiency along with reduced TIR ends. Second, the benefits of bbYY1 for precise TIR targeting may be sophisticated *in vivo*, and such complicated conditions cannot be represented in the *in vitro* cleavage assays. Third, published articles have shown that the RAG1 and RAG2 complexes target their target DNA via a dynamic movement and allosteric changes during the entire recombination and transposition progress^{3, 4, 5, 6}. Since bbYY1 can interact with bbRAG2, we think that such an interaction may influence the conformation of the TIR-bbRAGL complex, which may not be fully represented in the *in vitro* cleavage assays. Fourth, as you have mentioned, the results from *in vitro* cleavage assays were not sufficient to explain the reduced transposition activity of *ProtoRAG* in the presence of bbYY1. Thus, the *in vitro* cleavage assays have been removed from this revised manuscript. We hope that you find our responses acceptable.

In addition, according to you and the 3rd reviewer's comments, we have reconstructed our manuscript to make it more logical and easily for the readers to follow. We also have received editing by English Experts from the "American Journal Experts" website. We highly appreciate your help in improving our manuscript.

Reviewer #3 (Remarks to the Author):

While this manuscript has somewhat improved, my central concern remains: it is overly long and instead of telling one consistent and clear story it tries to overwhelm both the reader and the reviewers with too much data (8 supplemental figures) – but as one could see from the detailed comments of reviewer #1 on the previous round, there were tons of little issues with almost everyone of them. And within all these experimental finding there is unfortunately still very little support for their main conclusion that “ ... BbYY1 contributes to host domestication of the protoRAG transposon ...”. There is no evidence anywhere that either the TIRs or BbYY1 evolved in a certain way to allow for these observed variety of “regulatory modes” (only TIRs of one ProtoRAG fragment are tested, and seemingly HsYY1 works just as well as BbYY1) – it may as well that both, the ProtoRAG transposon and BbYY1, have co-existed in their current form over more 100 millions of years. In the last round I suggested to focus on the “regulation” as a theme, but the authors still tried to push too much in the direction of “domestication”.

Response to Reviewer #3 (1)

We highly appreciated your constructive comments. To avoid misleading the readers, in this revised manuscript, we have focused more on the “regulation” but not on the “domestication” of *ProtoRAG*. We have also removed some extraneous data to focus the manuscript. For example, parallel studies of human YY1 in the regulation of *ProtoRAG* have been completely removed. We hope that you find our responses acceptable.

To illustrate, I pick three examples where even the revised version falls short reaching the bar:

A) My previous concern 9 and the author's response #33.

This manuscript is focused on understanding the biology of the ProtoRAG transposon in *B. belcheri*. So the relevant YY1 molecule to use for the studies is BbYY1. As no *B. belcheri* cell line exists, human 293T cells are used instead and the HsYY1 knock-down cell line reconstituted with BbYY1 is the appropriate testbed. Why is it important to use BbYY1? Because it differs in the central feature from HsYY1: it lacks the classical activation domain. So the rationale behind reporting lots of experiments with HsYY1 while an appropriate cell model with BbYY1 was available is unclear to me. Omitting some of this superfluous data would make the story much easier to understand.

Response to Reviewer #3 (2)

Thank you for your constructive suggestions. According to your suggestions, most data regarding HsYY1 have been removed from the revised manuscript.

B) My previous concerns 4 and the author's response #28.

The rationale behind studying the HDJ formation (and even more so why this is done right after studying a role of BbYY1 in TIR-dependent DNA recombination) remains completely unclear – would one not first want to study transposition (as this is a transposon after all) when trying to understand transposon biology? The importance of HDJ in both *B. belcheri* biology and ProtoRAG transposon activity is unclear, the only point where it makes sense to study it is when trying to reveal similarities and differences to vertebrate RAGs.

Response to Reviewer #3 (3)

Thank you for your constructive comments. As we have shown in the manuscript, YY1 can bind to the TR5 element of *ProtoRAG*, which is important for HDJ formation in cells. Thus, when we observed that YY1 could target TR5, it was natural for us to think about whether YY1 could have some role in the formation of HDJs. Moreover, although *ProtoRAG* is a

self-transposon, HDJ formation is an important procedure for maintaining host genome stability. Thus, at the beginning of this study, we thought that testing the effects of bbYY1 on HDJ formation might provide some important hints to study the roles of bbYY1 in the activity of *ProtoRAG*. In fact, we did find that bbYY1 benefited precise TIR targeting and subsequent DNA rejoining, which might affect the transposition activity of *ProtoRAG*. We also found that bbYY1 could suppress the transposition of *ProtoRAG* and improve TIR-TIR rejoining after transposon excision. Thus, bbYY1 is a host factor that not only represses the transposition activity of *ProtoRAG* but also benefits host DNA rejoining to maintain genome stability. Although we observed some interesting phenomena in 293T cells, the protein domain architecture of bbYY1 is not fully conserved from amphioxus to humans, which is a phenomenon we would like to further address. Thus, the intact roles of bbYY1 *in vivo* still require further study using bbYY1 KO larva or cells, which are not fully established at the current stage. According to your comments, we have added a further explanation of the necessity of studying HDJ in the revised manuscript. We hope that you find our responses acceptable.

C) My previous concern 2 and the author's response #26.

Now that the authors state they also found other TF-binding sites (new here compared to the previous version). When trying to understand the biology of a transposon would one not start with the big effects (the 5'TIR boosts transcription 120-fold over background), yet here this study is centered on the two-fold decrease to what would be 60-fold (Figure 1) in transcriptional stimulation. One must simply wonder why the other factors were ignored (despite no-perfect matches of their binding sites) and whether those may have even bigger effects of *ProtoRAG* transposon behavior at all levels ranging from TIR recognition to transposition. I could not comment on this in the previous version as I did not know these facts. Now it makes me wonder whether these other factors are overall more important.

Response to Reviewer #3 (4)

Thank you for your questions. As you have pointed out, there are some other transcriptional factors that were found to bind the core *cis*-element of *ProtoRAG*. According to the prediction using the JASPAR program, YY1 received the highest prediction score by binding to the TR5 element within both 5' and 3' TIRs (See the Supplementary Table 1). To make the study of bbYY1 more logical, in this round of revision, certain *trans* factors with high match scores by binding to the TIR of *ProtoRAG* were used to test their effects on the transcription of *ProtoRAG*. As shown in Fig. 1h, among all these tested *trans* factors, YY1 had the most significant effect on reducing the transcription of *ProtoRAG*. At the recent stage, it is difficult for us to discuss the roles of these *trans* factors in the transcription of *ProtoRAG*, but the regulation of *ProtoRAG* is clearly complicated *in vivo* and requires further study. We very much appreciate your constructive comments for improving our manuscript.

References

1. Tao X, *et al.* Functional requirement of terminal inverted repeats for efficient ProtoRAG activity reveals the early evolution of V(D)J recombination. *Natl. Sci. Rev.* **7**, 403-417 (2019).
2. Zhang Y, *et al.* Transposon molecular domestication and the evolution of the RAG recombinase. *Nature* **569**, 79-84 (2019).
3. Ru H, Chambers MG, Fu TM, Tong AB, Liao M, Wu H. Molecular Mechanism of V(D)J Recombination from Synaptic RAG1-RAG2 Complex Structures. *Cell* **163**, 1138-1152 (2015).
4. Kim MS, *et al.* Cracking the DNA Code for V(D)J Recombination. *Mol. Cell* **70**, 358-370.e354 (2018).
5. Chen X, Cui Y, Wang H, Zhou ZH, Gellert M, Yang W. How mouse RAG recombinase avoids DNA transposition. *Nat. Struct. Mol. Biol.* **27**, 127-133 (2020).
6. Ru H, *et al.* DNA melting initiates the RAG catalytic pathway. *Nat. Struct. Mol. Biol.* **25**,

732-742 (2018).

REVIEWERS' COMMENTS:

Reviewer #2 (Remarks to the Author):

I have one minor comment:

Pg 17, lines 319-321. "bbYY1 expression vector or control vector were then co-transformed..."

However Fig 6A does not obviously show any conditions in which a control vector is used. Moreover, why is Fig 6A referred to before Fig 5?

Reviewer #3 (Remarks to the Author):

This manuscript has improved in terms of clarity and is now significantly easier to read. The revised title is a better choice compared to the previous one. The only concern remaining (that I think should be resolved in a future study) is the identity of additional critical trans-acting factors that regulate the transcription by binding to the TIRs. Specifically, it seems quite counterintuitive that all factors tested thus far, have some suppressive effect and it still makes me wonder a lot which factors are actually responsible for the positive regulation of transcription. In this context, the observed effects (2-fold suppression by BbYY1) still seem small compared to the overall 120-fold increase of transcription compared to the background.

All other concerns are now discussed in some detail in the manuscript and will help the readers to critically evaluate the data by themselves.

Point-by-point responses to the referees' comments

REVIEWERS' COMMENTS:

Reviewer #2 (Remarks to the Author):

I have one minor comment:

Pg 17, lines 319-321. "bbYY1 expression vector or control vector were then co-transformed..."
However Fig 6A does not obviously show any conditions in which a control vector is used.
Moreover, why is Fig 6A referred to before Fig 5?

Response to Reviewer #2:

Thank you very much for pointing out this misleading description. We have corrected it in the revised manuscript. Concerning your question about the Fig 6a citation order in the text, we have adjusted the relevant figure panels to make sure that all figures are cited in the order in the text. Thank you again for your help in improving this manuscript.

Reviewer #3 (Remarks to the Author):

This manuscript has improved in terms of clarity and is now significantly easier to read. The revised title is a better choice compared to the previous one. The only concern remaining (that I think should be resolved in a future study) is the identity of additional critical trans-acting factors that regulate the transcription by binding to the TIRs. Specifically, it seems quite counterintuitive that all factors tested thus far, have some suppressive effect and it still makes me wonder a lot which factors are actually responsible for the positive regulation of transcription. In this context, the observed effects (2-fold suppression by BbYY1) still seem small compared to the overall 120-fold increase of transcription compared to the background. All other concerns are now discussed in some detail in the manuscript and will help the readers to critically evaluate the data by themselves.

Response to Reviewer #3:

We highly appreciated to your constructive comments on the manuscript improvement.